# A *Cognac* Shot To Forget Bad Memories:
# Corrective Unlearning for Graph Neural Networks

**Varshita Kolipaka** [* 1]   **Akshit Sinha** [* 1]   **Debangan Mishra** [1]   **Sumit Kumar** [1]
**Arvindh Arun** [† 1 2]   **Shashwat Goel** [† 1 3 4]   **Ponnurangam Kumaraguru** [1]

🌐 cognac-gnn-unlearning.github.io   💠 corrective-unlearning-for-gnns

## Abstract

Graph Neural Networks (GNNs) are increasingly being used for a variety of ML applications on graph data. Because graph data does not follow the independently and identically distributed (*i.i.d.*) assumption, adversarial manipulations or incorrect data can propagate to other data points through message passing, which deteriorates the model's performance. To allow model developers to remove the adverse effects of manipulated entities from a trained GNN, we study the recently formulated problem of *Corrective Unlearning*. We find that current graph unlearning methods fail to unlearn the effect of manipulations even when the whole manipulated set is known. We introduce a new graph unlearning method, **_Cognac_**, which can unlearn the effect of the manipulation set even when only 5% of it is identified. It recovers most of the performance of a strong oracle with fully corrected training data, even beating retraining from scratch without the deletion set, and is 8x more efficient while also scaling to large datasets. We hope our work assists GNN developers in mitigating harmful effects caused by issues in real-world data, post-training.

## 1. Introduction

Graph Neural Networks (GNNs) are seeing widespread adoption across diverse domains, from recommender systems to drug discovery (Wu et al., 2022; Zhang et al., 2022). Recently, GNNs have been scaled to large training sets for

various graph foundation models (Mao et al., 2024; Arun et al., 2025). However, in these large-scale settings, it is prohibitively expensive to verify the integrity of every sample in the training data that can potentially affect desiderata like fairness (Konstantinov & Lampert, 2022), robustness (Paleka & Sanyal, 2023; Günnemann, 2022), and accuracy (Sanyal et al., 2021).

Making the training process robust to minority populations (Günnemann, 2022; Jin et al., 2020) is challenging and can adversely affect fairness and accuracy (Sanyal et al., 2022). Consequently, model developers may want post-hoc ways to remove the adverse impact of manipulated training data if they observe problematic model behavior on specific distributions of test-time inputs. Such an approach follows the recent trend of using post-training interventions to ensure models behave in intended ways (Ouyang et al., 2022). Recently, Goel et al. (2024) formulated corrective unlearning as the challenge of removing adverse effects of manipulated data with access to only a representative subset for unlearning while being agnostic to the type of manipulations. We study this problem in the context of GNNs, which face unique challenges due to the graph structure. The traditional assumption of independent and identically distributed (*i.i.d.*) samples does not hold for GNNs, as they use a message-passing mechanism that aggregates information from neighbors. This process makes GNNs vulnerable to adversarial perturbations, where modifying even a few nodes can propagate changes across large portions of the graph and result in widespread changes in model predictions (Bojchevski & Günnemann, 2019b; Zügner et al., 2018). Consequently, for GNNs to effectively unlearn, they must remove the influence of manipulated entities on their neighbors.

*Corrective Unlearning* is an emerging paradigm that focuses on removing the influence of arbitrary training data manipulations on a trained model, using only a representative subset of the manipulated data (Goel et al., 2024). In this work, we focus on the use of GNNs in node classification tasks, studying unlearning for targeted binary class confusion attacks (Lingam et al., 2024) on both edges and nodes. For edge unlearning, we evaluate the unlearning of spurious

---

[*]Equal contribution [†]Equal advising. [1]IIIT Hyderabad [2]Institute for AI, University of Stuttgart [3]ELLIS Institute Tübingen [4]Max Planck Institute for Intelligent Systems. Correspondence to: Varshita Kolipaka <varshita.k@research.iiit.ac.in>, Akshit Sinha <akshit.sinha@students.iiit.ac.in>.

*Proceedings of the 42ⁿᵈ International Conference on Machine Learning*, Vancouver, Canada. PMLR 267, 2025. Copyright 2025 by the author(s).

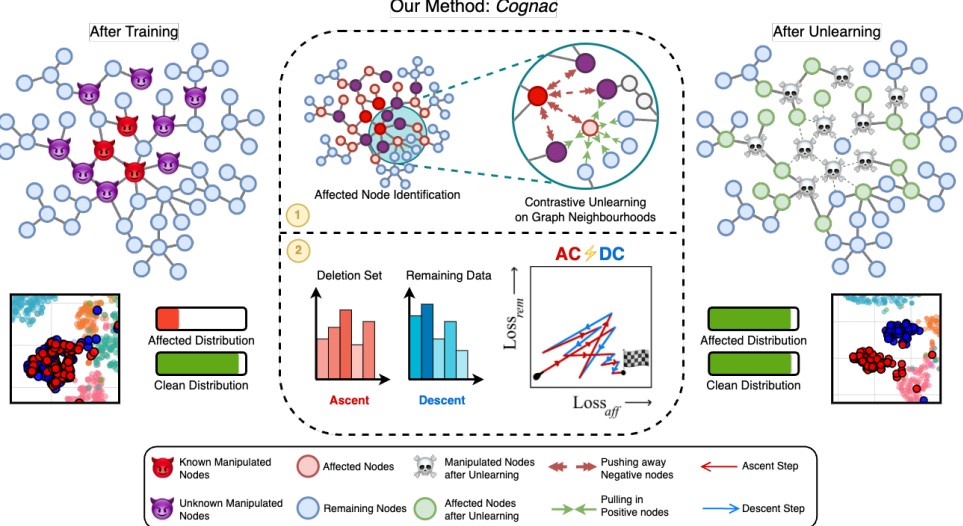

*Figure 1.* **Illustration of our method *Cognac*.** Initially (Left), the model is trained on manipulated data (Devils), out of which only a subset is identified for deletion (Dark-red-devils). Our method alternates between two steps. (1) Identifying neighbors by the deletion set, which can include both nodes from the remaining data (light red) and unidentified manipulated nodes (Purple), and pushes their representation away from the deletion set and toward other nodes in the neighborhood. (2) We then perform ascent on the deletion set labels and descent on the remaining data with separate optimizer instances. This cleanly separates the embeddings of the affected classes (Right), recovering the accuracy on the affected distribution, and maintaining it on the remaining distribution.

edges that change the graph topology in a way that violates the homophily assumption that most GNNs rely on. For node unlearning, we utilize a label flip attack (Lingam et al., 2024) which is used as a classical graph adversarial attack, similar to the Interclass Confusion attack (Goel et al., 2022).

First, we evaluate whether existing GNN unlearning methods are effective in removing the impact of manipulated entities. Our findings reveal that these methods consistently fail, even when provided with a complete set of manipulated entities. We then propose our method, ***Cognac***, which unlearns by alternating between two components, as illustrated in Figure 1. The first component *Contrastive unlearning on Graph Neighborhoods* (***CoGN***), finds affected neighbors of the known deletion set, updating the GNN weights using a contrastive loss that pushes representations of the affected neighbors away from the deletion entities while staying close to other neighbors. The second component, *AsCent DesCent de♮coupled* (***AC♮DC***) applies the classic *i.i.d.* unlearning method of gradient ascent on the *deletion set* and gradient descent on the *retain set*.

Our proposed method shows promise for corrective unlearning: we not only outperform retraining from scratch, the previously assumed gold standard for this task, but also recover most of the performance of an oracle (a model trained on the complete and correct data) while discovering as few as 5% of the manipulated entities.

## 2. Corrective Unlearning for Graph Neural Networks

We now formulate the *Corrective Unlearning* problem for graph-structured, non-*i.i.d.* data. We consider a graph $\mathcal{G} = (\mathcal{V}, \mathcal{E})$, where $\mathcal{V}$ and $\mathcal{E}$ represent the constituent set of nodes and edges respectively. For each node $\mathcal{V}_i \in \mathcal{V}$, there is a corresponding feature vector $\mathcal{X}_i$ and label $\mathcal{Y}_i$, with $\mathcal{V} = (\mathcal{X}, \mathcal{Y})$. Consistent with prior work in unlearning on graphs (Wu et al., 2023a; Li et al., 2024c), we focus on semi-supervised node classification using GNNs. GNNs use the message-passing mechanism, where each node aggregates features from its immediate neighbors. The effect of this aggregation process propagates through multiple successive layers, effectively expanding the receptive field of each node with network depth. This architecture inherently exploits the principle of homophily, a common property in many real-world graphs where nodes with similar features or labels are more likely to be connected than not.

While assuming homophily is extremely useful for learning representations from graph data, annotation mistakes or adversarial manipulations that create dissimilar neighborhoods or connect otherwise dissimilar nodes can easily harm the learned representations (Zügner & Günnemann, 2019). This motivates our study of post-hoc correction strategies like unlearning for GNNs. Following Goel et al. (2024), we adopt an adversarial formulation that subsumes correcting more benign mistakes.

**Adversary's Perspective.** The adversary aims to reduce model accuracy on a target distribution by manipulating parts of the clean training data $\mathcal{G}$. This can be done in the following ways: (1) adding spurious edges $\hat{\mathcal{E}}$, resulting in $\mathcal{E}' = \mathcal{E} \cup \hat{\mathcal{E}}$; or (2) manipulating node information, $\mathcal{V}' = f_m(\mathcal{V})$, where $f_m$ manipulates a subset of nodes by changing their features or labels. We define $S_m$ as the set of manipulated entities, which can be either the manipulated subset of nodes or the added spurious edges $\hat{\mathcal{E}}$. The final manipulated graph is denoted as $\mathcal{G}' = (\mathcal{V}', \mathcal{E}')$.

**Unlearner's Perspective.** After training, model developers may observe that desired properties like fairness and robustness are compromised in the trained model $\mathcal{M}$, which can be modeled as lower accuracy on some data distributions. The objective, then, is to remove the influence of the manipulated training data $S_m$ on the affected distribution while maintaining performance on the remaining entities. By utilizing data monitoring strategies on a subset of the training data or using incorrect data detection techniques like (Northcutt et al., 2021), it may be possible to identify a part of the manipulated entities $S_f \subseteq S_m$. For unlearning to be feasible, $S_f$ must be a representative subset of $S_m$. We only assume the type of affected entity (edges or nodes) is known to the model developer, but do not assume any knowledge about the nature of manipulation. An unlearning method $U(\mathcal{M}, S_f, \mathcal{G}')$ is then used to mitigate the adverse effects of $S_m$, ideally by improving the accuracy on unseen samples from the affected distribution. An effective unlearning method should remove the impact of certain training data samples without degrading performance on the rest of the data or incurring the cost of retraining from scratch. Moreover, while Retrain was previously considered a gold standard in privacy-oriented unlearning and graph unlearning, Goel et al. (2024) showed that when the whole manipulated set is not known, retraining on the remaining data can reinforce the manipulation, implying it's not a gold standard for corrective unlearning.

**Metrics.** To evaluate the performance of unlearning methods, we use the metrics proposed by Goel et al. (2024):

1. **$\text{Acc}_{\text{aff}}$**: It measures the clean-label accuracy of test set samples from the affected distribution. This metric captures the method's ability to *correct* the influence of the manipulated entities on unseen data through unlearning. As the affected distribution differs for each manipulation, we specify it when describing each evaluation.

2. **$\text{Acc}_{\text{rem}}$**: It is defined as the accuracy of the remaining entities. This metric measures whether the unlearning maintains model performance on clean entities.

The metrics $\text{Acc}_{\text{aff}}$ and $\text{Acc}_{\text{rem}}$ were termed "Corrected Accuracy" ($\text{Acc}_{\text{corr}}$) and "Retain Accuracy" ($\text{Acc}_{\text{retain}}$) respectively by Goel et al. (2024). We chose alternative names

to explicitly state which data distribution accuracy is measured. In Section 4, we further specify what the "affected distribution" and "remaining entities" are for the different evaluation types we study.

**Goal.** An ideal corrective unlearning method should have high $\text{Acc}_{\text{aff}}$ even when a small fraction of manipulated set ($S_m$) is identified for deletion ($S_f$) without big drops in $\text{Acc}_{\text{rem}}$, all while being computationally efficient.

## 3. Our Method: *Cognac*

Our proposed unlearning method, *Cognac*, requires access to the underlying graph $\mathcal{G}'$, the known set of entities to be deleted $S_f$, and the original model $\mathcal{M}$. We define $\mathcal{V}_f$ as the set of nodes whose influence is to be removed. For node unlearning, $\mathcal{V}_f = S_f$; for edge unlearning, $\mathcal{V}_f$ is the set of vertices connected to the edge set to be deleted. Manipulated data has two main adverse effects on the trained GNN: 1) Message passing can propagate the influence of the manipulated entities $S_m$ on their neighborhood, and 2) The layers learn transformations to fit potentially wrong labels in $S_m$. Mitigating the effects of attacks first requires analysis of the impact of such attacks. The most general form of attack is Interclass Confusion (Goel et al., 2022), which we show entangles the representations of two classes, below.

**Theorem 3.1.** *Let $G = (V, E, X)$ be a graph with node set $V$, edge set $E$, and features $X$. Let $C_1, C_2 \subset V$ be two distinct classes with ground-truth labels $y_i \in \{C_1, C_2\}$. Suppose an Interclass Confusion (IC) attack is applied.*

*Let $\phi_M(C_1), \phi_M(C_2) \in \mathbb{R}^d$ denote the mean embeddings of $C_1$ and $C_2$, and $\mathcal{D}(\phi_M(C_1), \phi_M(C_2))$ be the Wasserstein-2 distance between their embedding distributions.*

*Then, there exists a degradation term $\Delta > 0$ such that:*

$$\mathbb{E}\left[\mathcal{D}(\phi_M(C_1), \phi_M(C_2))\right]$$
$$\leq \mathbb{E}\left[\mathcal{D}(\phi_{M_{clean}}(C_1), \phi_{M_{clean}}(C_2))\right] - \Delta$$

Complete list of assumptions and attack details in Appendix A.3. We tackle these two problems using separate components - CoGN and AC⚡DC.

### 3.1. Removing Effects on Neighbors with CoGN

The first question we address is: *How can we remove the influence of manipulated entities on their neighboring nodes?* First, this requires us to identify the nodes affected by the manipulations ($\mathcal{V}_{\text{aff}}$) and then mitigate the influence on their representations. Identifying affected nodes is challenging, as the impact of message passing from manipulated entities $S_m$ depends on the interference from messages of other neighboring nodes. Therefore, we use an empirical estimation to identify the affected nodes from each entity in the

deletion set. On these nodes, we then perform contrastive unlearning, simultaneously pushing the representations of the affected nodes away from nodes in $\mathcal{V}_f$ while keeping them close to other nodes in their neighborhood. We call this component *Contrastive unlearning on Graph Neighborhoods* (**CoGN**), formalized below.

### 3.1.1. AFFECTED NODE IDENTIFICATION

To first identify the affected samples, $\mathcal{V}_{\text{aff}}$, we first observe that any manipulated node $s \in V_f$ can only affect the representation of any other node $v \in \mathcal{V}' \backslash V_f$ in $\mathcal{G}'$ only if $s$ is part of the *receptive field* of $v$, a widely known result for GNNS. Formally,

**Lemma 3.2.** *Let $G = (V, E)$ be an undirected graph, and let $\mathcal{N}(v)$ denote the 1-hop neighborhood of node $v$. For a node $s \in V$, let $\mathcal{N}^n(s)$ denote the $n$-hop neighborhood of $s$, defined recursively as:*

$$\mathcal{N}^n(s) = \begin{cases} \{s\} & \text{if } n = 0, \\ \bigcup_{v \in \mathcal{N}^{n-1}(s)} \mathcal{N}(v) & \text{if } n > 0. \end{cases}$$

*In an $n$-layer GNN, the representation $z_s$ of node $s$ can affect the representations $z_v$ of nodes $v$ only if $v \in \mathcal{N}^n(s)$. For any $v \notin \mathcal{N}^n(s)$, $z_v$ is independent of message passing effects of $z_s$.*

The proof is provided in Appendix A.1. The main takeaway is that manipulations only propagate within an $n$-hop neighborhood of poisoned nodes. This locality drastically reduces the search space for identifying affected nodes.

The second observation is that not all nodes in the $n$-hop neighborhood of the manipulated nodes may be affected enough by the attack, as some nodes are more robust to the perturbations than others (Gosch et al., 2023; Arun et al., 2023). To find the most affected nodes, we employ a cheap and simple heuristic. We invert the features of $v \in V_f$ and select neighboring nodes where final output logits are changed the most. Formally, the inversion is performed by the transformation $\vec{1} - \mathcal{X}_v, \forall v \in \mathcal{V}_f$, leading to a new feature matrix $\chi'$, where $\mathcal{X}_v$ represents a one-hot-encoding vector. We then compute the difference in the original output logits $\mathcal{M}(\chi)$, and those obtained by on the new feature matrix, $\mathcal{M}(\chi')$ given by: $\Delta\chi = |\mathcal{M}(\chi') - \mathcal{M}(\chi)|$. The top $k\%$ nodes with the most change, $\Delta\chi$, are selected as the affected set of entities $\mathcal{V}_{\text{aff}}$. A conceptual example illustrating the workings of Affected Node Identification is presented in Figure 2. Appendix E.2 varies our design choices, confirming that we retain the same performance as using the entire $n$-hop neighborhood while being more efficient (Figure 9). Our method works robustly even if the original GNN was under-trained (Figure 8). Further, in Table 7 we also ablate the heuristic for identifying affected nodes against *Cognac* using MEGU's sampling technique. We observe that our heuristic delivers over 25% higher and is 8x faster.

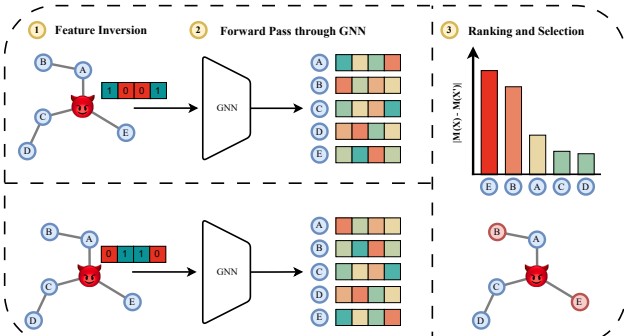

*Figure 2.* **A toy example detailing Affected Node Identification.** (1) We first invert the features of the known manipulated node. (2) We perform two forward passes through the GNN, one with the original feature vector and one with the inverted feature vector. (3) We compute the difference in output logits from both cases and take the top 2 nodes with the largest logit change, which in this case are nodes E and B.

### 3.1.2. CONTRASTIVE UNLEARNING

To remove the influence of the deletion set $\mathcal{V}_f$ on the affected nodes $\mathcal{V}_{\text{aff}}$ identified in the previous step, we must ensure that our model maps the hidden representations of nodes in $\mathcal{V}_{\text{aff}}$ far away from that of nodes in $\mathcal{V}_f$ (Goel et al., 2024). However, satisfying this property alone will lead to unrestricted separation and damage the quality of learned representations. To achieve this balance and restore homophily (Ma et al., 2022), we constrain $\mathcal{V}_{\text{aff}}$ to stay close to its unaffected neighbors $\mathcal{N}_{\mathcal{V}_{\text{aff}}} \backslash \mathcal{V}_f = \mathcal{V}_{pos}$, ensuring alignment with the local structure while distancing from $\mathcal{V}_f$. Formally, we can state this as the following optimization problem over the parameters of a GNN, which is necessary for corrective unlearning,

$$\max_\theta \left( \mathbb{E}_{v \in \mathcal{V}_{\text{aff}}, p \in \mathcal{V}_{pos}} \left[ \mathbf{z}_v^\top \mathbf{z}_p \right] - \mathbb{E}_{v \in \mathcal{V}_{\text{aff}}, n \in \mathcal{V}_f} \left[ \mathbf{z}_v^\top \mathbf{z}_n \right] \right)$$

$$= \max_\theta \left( \mathbb{E}_{v \in \mathcal{V}_{\text{aff}}, p \in \mathcal{V}_{pos}, n \in \mathcal{V}_f} \left[ \mathbf{z}_v^\top \mathbf{z}_p - \mathbf{z}_v^\top \mathbf{z}_n \right] \right) \quad (1)$$

Equation 1 offers a direct way to enforce separation between positive and negative pairs, but it has notable shortcomings. In particular, it does not sufficiently penalize small margins, as it only considers the raw similarity difference without emphasizing cases where $z_v^T z_p$ and $z_v^T z_n$ are nearly equal. This can lead to weak separation and reduced robustness, especially when positive and negative embeddings are closely aligned. Additionally, the lack of non-linear scaling creates an uneven optimization landscape, increasing the risk of convergence issues. To overcome these limitations, we use a log-based loss function that applies non-linear sigmoid terms, providing stronger penalization for small margins and a probabilistic similarity interpretation (Hamilton et al., 2017). This enhances the separation between positive and negative pairs while ensuring smoother gradients for more stable optimization (Lee et al., 2024). Formally,

$$\mathcal{L}_{v,p,n} = -\log(\sigma(z_v^T z_p)) - \log(\sigma(-z_v^T z_n)) \quad (2)$$

Under assumptions of differentiability, convexity, and bounded gradients, we can guarantee that optimizing this function achieves a better separation between the positive and negative dot products equivalent to Equation 1.

**Theorem 3.3.** *Let $\theta \in \mathbb{R}^d$ parameterize a GNN generating embeddings $z_v, z_p, z_n \in \mathbb{R}^k$ for triplets $(v, p, n)$. Let $\mathcal{L}(\theta) = -\mathbb{E}\left[\log \sigma(z_v^\top z_p) + \log \sigma(-z_v^\top z_n)\right]$ and $S(\theta) = \mathbb{E}\left[z_v^\top z_p - z_v^\top z_n\right]$.*

*Then under the assumptions stated above, gradient descent on $\mathcal{L}(\theta)$ with step size $\eta \leq \frac{1}{L}$ (where $L$ is the Lipschitz constant of $\nabla_\theta \mathcal{L}$) guarantees:*

$$S(\theta_{t+1}) \geq S(\theta_t) \quad \forall t \geq 0,$$

*and at convergence, $S(\theta^*) > S(\theta_0)$.*

The proof is presented in Appendix A.2. We also empirically validate this theorem for non-linear GNNs, with Figure 3 showing on the *Cora* dataset that the separation between the positive dot product and negative dot product monotonically increases as the loss converges.

### 3.2. Unlearning Old Labels with AC♮DC

Next, we ask: *Can we undo the effect of the task loss $\mathcal{L}_{\text{task}}$ explicitly learning to fit the node representations of $S_m$ to potentially wrong labels?* We do this by performing gradient ascent on $S_f$, which non-directionally maximizes the training loss with respect to the old labels. Ascent alone aggressively leads to arbitrary forgetting of useful information, so we counterbalance it by alternating with steps that minimize the task loss on the remaining data. More precisely, we perform gradient ascent on $\mathcal{V}_f$ and gradient descent on $\mathcal{V}_r$, iteratively on the original GNN $\mathcal{M}$.

$$\mathcal{L}_a = -\mathcal{L}_{\text{task}}(\mathcal{V}_f), \; \mathcal{L}_d = \mathcal{L}_{\text{task}}(\mathcal{V}_r) \qquad (3)$$

While variants of ascent on $S_f$ and descent on remaining data have been studied for image classification (Kurmanji et al., 2023) and language models (Yao et al., 2024), we find a specific optimization strategy useful to achieve corrective unlearning on graphs. The challenge arises when $S_f \subset S_m$, as the remaining data could still contain manipulated entities, which we aim to avoid reinforcing. However, in realistic scenarios, the manipulated entities $S_m$ typically are a small fraction of the training data, allowing us to mitigate their impact through ascent on the representative subset $S_f$.

This requires a careful balance between ascent and descent, which we can achieve by using two different optimizers and starting learning rates for these steps. This insight is similar to prior work in Generative Adversarial Networks (GANs) (Heusel et al., 2017). The starting learning rates for both ascent and descent are hyperparameters to be tuned, and usually, we find that a lower learning rate for ascent

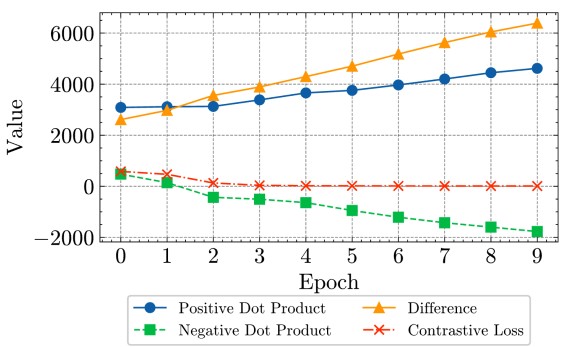

*Figure 3.* **Empirical convergence of CoGN.** The average positive and negative dot products across samples increase and decrease, respectively, over epochs, resulting in overall convergence.

leads to better results. Thus, we call this component *Ascent Descent de♮coupled*. Convergence of this formulation has been shown in previous works (Kurmanji et al., 2023). We show the empirical convergence of AC♮DC in Figure 7 present in Appendix E.1.

For our final method ***Cognac***, we alternate steps of *CoGN*, which fixes representations of affected neighborhood nodes, and *AC♮DC*, which unlearns potentially wrong labels introduced by $S_m$. We also perform ablations in Section 5 (Table 3), showing that the individual components CoGN and AC♮DC alone perform notably worse than our final method, suggesting the necessity of both components.

## 4. Experimental Setup

### 4.1. Benchmarking Details

We now describe design choices made for benchmarking, first specifying the datasets and architectures, and then outlining how to ensure a fair comparison between methods.

**Models and Datasets.** We report results using the Graph Convolutional Network (GCN) (Kipf & Welling, 2017) architecture and evaluate the methods on ten benchmark datasets used in prior literature: Cora, PubMed, DBLP, Coauthor CS, Coauthor Physics, Amazon Photos (Photos), Amazon Computers (Computers), CiteSeer (Cheng et al., 2023; Li et al., 2024c), and additionally CoraFull (Bojchevski & Günnemann, 2018) and OGB-arXiv (Wang et al., 2020). In Appendix D.1, we provide additional results on Graph Attention Network (GAT) (Veličković et al., 2018). For each dataset, we extract the largest connected subgraph for our experiments. Dataset details, including the number of classes, nodes, edges, and entities manipulated, are provided in Table 1 and Appendix B.

**Hyperparameter Tuning.** Ensuring a fair comparison of unlearning methods can be tricky, as there are multiple desiderata: unlearning, maintaining utility, and computa-

*Table 1.* **Dataset statistics**. In our evaluations, we include citation networks, co-author networks, and co-purchase networks. The number of nodes and edges reported here refers to the entire dataset. From this, we use a 60/20/20 split for train/validation/test. The manipulation statistics are presented in Table 4.

| DATASET | CLASSES | NODES | EDGES |
|---|---|---|---|
| CORAFULL | 70 | 18, 800 | 125, 370 |
| CORA | 7 | 2, 485 | 10, 138 |
| CITESEER | 6 | 2, 120 | 7, 358 |
| DBLP | 4 | 16, 191 | 103, 826 |
| PUBMED | 3 | 19, 717 | 88, 648 |
| OGB-ARXIV | 40 | 169, 343 | 1, 166, 243 |
| CS | 15 | 18, 333 | 163, 788 |
| PHYSICS | 5 | 34, 493 | 495, 924 |
| PHOTOS | 8 | 7, 487 | 238, 086 |
| COMPUTERS | 10 | 13, 381 | 491, 556 |

tional efficiency, and hyperparameter tuning of the methods can particularly affect results on GNNs. We describe our efforts towards this in Appendix sections F.1 and F.2.

## 4.2. Evaluations

Given a fixed budget of samples to manipulate, ideal corrective unlearning evaluations should maximize the deterioration of model performance on the affected distribution, creating a wide gap between clean and poisoned model performance to measure the progress of the unlearning method. We thus evaluate unlearning in the context of attacks that are not constrained by stealthiness. Lingam et al. (2024) show that binary label flip manipulation attacks, where a fraction of labels are swapped between two chosen classes, are stronger than multi-class manipulations, theoretically and empirically, on GNNs. Building on this, we use two targeted attacks to evaluate corrective unlearning on graph data. Additionally, for completeness we also present results on a Feature Poisoning Attack, analogous to patch-based backdoor attacks in image poisoning (Gu et al., 2017), in Appendix D.3.

**Spurious Edge Addition.** Prior GNN unlearning works (Wu et al., 2023a; Li et al., 2024c) have evaluated adversarial edge attacks, but in an untargeted setting, making their evaluations weak. We, instead, simulate targeted adversarial edge insertions between nodes of two classes, violating homophily assumptions and entangling their representations. This models attacks like fake social connections (Bojchevski & Günnemann, 2019a) or knowledge graph manipulations (Xi et al., 2023; Zhang et al., 2019; Zhao et al., 2024). Unlearning aims to recover accuracy on the targeted classes $\text{Acc}_{\text{aff}}$ while preserving performance on others $\text{Acc}_{\text{rem}}$.

**Label Manipulation.** We implement the *Interclass Confusion (IC) Test* (Goel et al., 2022), by systematically swap-

ping labels between two targeted classes to entangle their representations. Once again, the unlearning goal is to improve $\text{Acc}_{\text{aff}}$ on the two targeted classes, while preserving performance, $\text{Acc}_{\text{rem}}$, on the remaining classes.

## 4.3. Baselines

We evaluate four popular graph unlearning methods and adapt one popular *i.i.d.* unlearning method for graphs. For reference, we also report results for the Original model, Retrain, which trains a new model without $S_f$, and Finetune, which continues training the poisoned model on data without $S_f$ for additional epochs. Following Goel et al. (2024), we find that retraining from scratch is not the gold standard in corrective unlearning. Thus, we introduce Oracle, trained on the whole training set without manipulations, indicating an upper bound on what can be achieved. The Oracle has correct labels for the unlearning entities, information that the unlearning methods cannot access.

**Existing Unlearning Methods.** We choose five methods as baselines where unlearning incorrect data explicitly motivates the technique. (1) *GNNDelete* (Cheng et al., 2023) adds a deletion operator after each GNN layer and trains them using a loss function to randomize the prediction probabilities of deleted edges while preserving their local neighborhood representation, keeping the original GNN weights unchanged. (2) *GIF* (Wu et al., 2023a) draws from a closed-form solution for linear GNNs to measure the structural influence of deleted entities on their neighbors. Then, they provide estimated GNN parameter changes for unlearning using the inverse Hessian of the loss function. (3) *MEGU* (Li et al., 2024c) finds the highly influenced neighborhood (HIN) of the unlearning entities and removes their influence over the HIN while maintaining predictive performance and forgetting the deletion set using a combination of losses. (4) *UtU* (Tan et al., 2024) proposes *zero-cost* edge-unlearning by removing the edges to be deleted during inference for blocking message propagation from nodes linked to these edges. Finally, we include a popular unlearning method studied in *i.i.d.* classification settings. (5) *SCRUB* (Kurmanji et al., 2023) employs a teacher-student framework with alternate steps of distillation away from the forget set and towards the retain set. For edge unlearning, we use SCRUB by taking the nodes connected to spuriously added edges as the forget set and the rest as the retain set.

## 5. Results & Discussion

We now report our main results comparing our method to existing methods across the manipulation types and datasets. Detailed method ablations and analyses of what can be achieved in this setting are reported in Appendix E.4. We also present consistent results on the GAT architecture in Appendix D.1. We show the robustness of *Cognac* to large

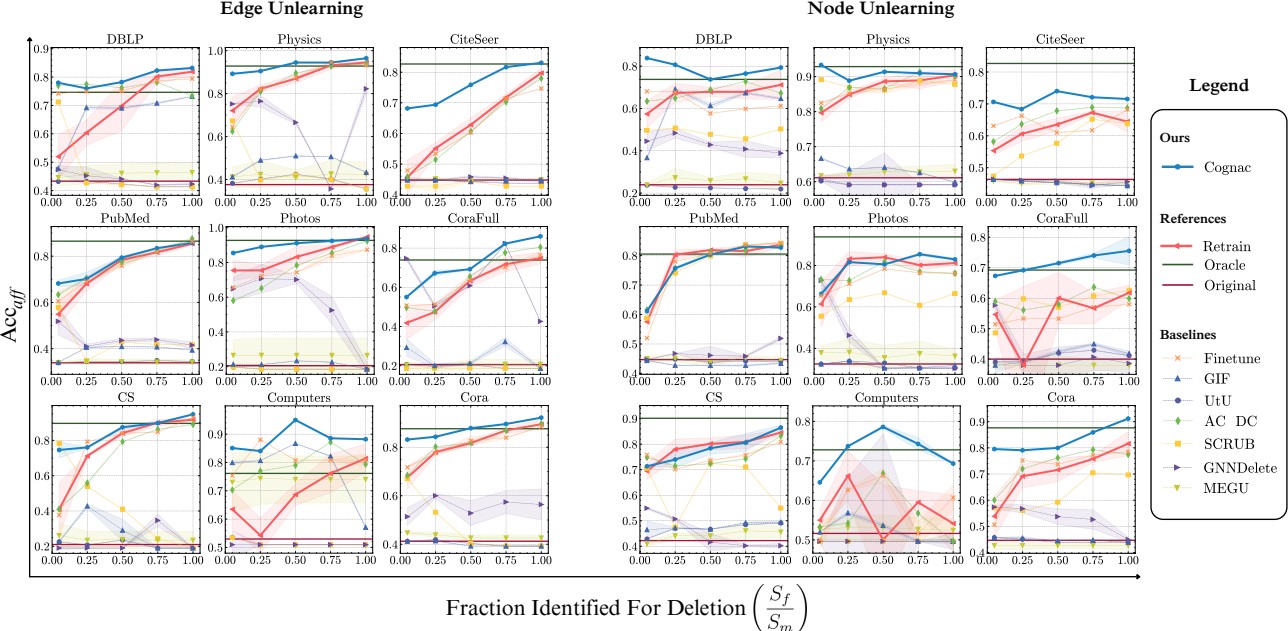

*Figure 4.* **Corrective Unlearning Results.** We report the accuracy on the affected classes $\text{Acc}_{\text{aff}}$ across different fractions of the manipulation set known for deletion ($S_f/S_m$). Baseline methods perform poorly, except for GNNDelete, which achieves reasonable unlearning performance in some settings. AC♮DC and Finetune, despite not being graph-specific, perform much better. *Cognac*, which adds graph awareness to AC♮DC, archives SOTA across datasets and corrective fractions, unlearning the effect of the manipulation with just 5% of the manipulation set known.

$S_f$ sizes, showing strong performance even with 38.96% of total training nodes in $S_f$ for the *PubMed* dataset.

*Table 2.* **Accuracy averaged across $S_f/S_m$ on remaining distribution relative to the Original model.** We find prior methods, especially GNNDelete, lead to large drops in $\text{Acc}_{\text{rem}}$, while our methods *Cognac* and AC♮DC minimize the loss in $\text{Acc}_{\text{rem}}$.

| METHOD | CS | | CORA | |
|---|---|---|---|---|
| | EDGE | LABEL | EDGE | LABEL |
| ORIGINAL | 90.6 | 89.6 | 61.2 | 61.4 |
| ORACLE | −0.1 | +0.5 | −0.7 | −3.0 |
| RETRAIN | −1.4 | −0.1 | −2.4 | −5.7 |
| *Cognac* | −1.4 | −0.4 | −4.5 | −2.9 |
| AC♮DC | −0.7 | −0.8 | −2.2 | −0.8 |
| GNNDELETE | −6.0 | −1.9 | −10.9 | −8.4 |
| GIF | −1.6 | −0.9 | −4.2 | −0.6 |
| MEGU | −0.5 | −2.8 | +0.0 | −6.5 |
| UTU | +0.0 | +0.0 | +0.0 | +0.0 |
| SCRUB | −0.9 | −0.7 | +0.0 | −4.8 |

Figure 4 shows unlearning performance on the test set for manipulated classes ($\text{Acc}_{\text{aff}}$) upon varying the fraction of the manipulation set known for unlearning ($S_f/S_m$). Table 2 accompanies this, showing side-effects on utility.

**1. Existing unlearning methods perform poorly even when $|S_f| = |S_m|$.** Observing the rightmost points in Fig-

ure 4, we can see across manipulation types and datasets that existing methods fail to improve $\text{Acc}_{\text{aff}}$ even when the whole manipulation set is known. UtU fails to unlearn the effects of either of the attacks, as simply unlinking on the forward pass does not sufficiently counteract the influence on neighbors and weights. Both SCRUB and MEGU use a KL Divergence Loss term to keep predictions on the remaining data close to the original model, which could be detrimental when done on unidentified manipulation set entities and other affected neighbors. Even though MEGU and GIF were evaluated on removing adversarial edges and GNNDelete also mentioned incorrect data as one of its key applications, they failed to recover performance when presented with targeted data manipulation. Interestingly, despite extensive hyperparameter searches, **they are beaten by methods with no special graph components**: the best baselines are AC♮DC and naive finetuning (Finetune) on the retain set: both achieve a performance similar to retraining in some of the evaluations.

**2. Corrective unlearning methods must perform better than Retrain.** While both AC♮DC and Finetune match Retrain, all of them still fall far behind the Oracle model's performance in most evaluations, and are not consistent enough. This scope for improvement motivates the design of our method which performs well across attacks, datasets, and fractions identified for deletion.

**3.** *Cognac* **beats Retrain, and can sometimes match the Oracle's performance.** We observe that *Cognac* consistently achieves state-of-the-art performance across all datasets and manipulations, convincingly and consistently beating existing graph methods, and often exceeds Retrain. Notably, *Cognac* occasionally even surpasses Oracle, our introduced gold standard, on multiple datasets and identified fractions - despite having access to less data (and unknown manipulated samples) than Oracle. In Table 3, we show how both components of *Cognac* complement each other. CoGN alone does not improve $Acc_{aff}$ over the original model, showing that while it effectively moves affected nodes, it lacks signals from labels. Conversely, $AC\natural DC$ alone achieves better performance than CoGN, but is still far from matching the Oracle. $AC\natural DC$ weakens incorrect learning signals and preserves task-relevant representations, while CoGN steers affected nodes away from manipulated ones. This highlights the effectiveness of our contrastive approach in achieving robust unlearning.

**4.** *Cognac* **performs strongly even with only 5% of $S_m$ known.** *Cognac* effectively recovers most of the accuracy on the affected distribution even when only 5% of the manipulated set is known. Notably, it outperforms Retrain in realistic scenarios where $S_m$ is only partially known. We attribute this to *Affected Neighborhood Identification*, which leverages graph structure to infer manipulated nodes and edges beyond those explicitly identified. Additionally, CoGN plays a crucial role by pushing influenced neighbors away from the deletion set and aligning them with their unaffected neighbors, thereby correcting representations even for unknown manipulated samples.

**5.** *Cognac* **scales well to significantly larger datasets.** We evaluate *Cognac* on OGB-Arxiv, a significantly larger dataset than our other benchmarks with $170,000$ nodes and over 1M edges, and observe that it continues to achieve strong performance. Despite the increased scale, *Cognac* maintains a substantial lead over Retrain across different fractions of $S_m$ (Figure 5 (a)), while the baseline methods

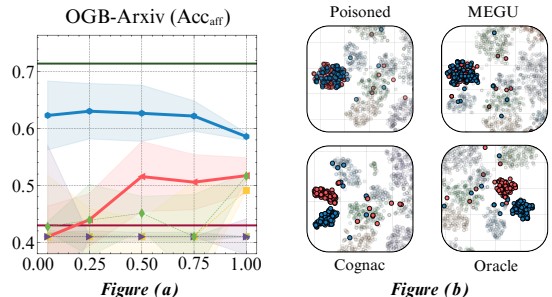

*Figure 5.* **(a) Results on a larger dataset, OGB-Arxiv, and (b) Visualization of hidden layer embeddings after unlearning on CS dataset for node unlearning**. (a) On the OGB-Arxiv dataset, *Cognac* outperforms retraining from scratch by more than 10%, while most baselines fail to achieve any performance gains beyond the Original model. (b) The affected distribution embeddings (highlighted by red and blue) are fully entangled in the original trained model, while after unlearning with *Cognac* the embeddings are well separated and clustered, matching Oracle.

fail to improve $Acc_{aff}$ over the original model. Even as dataset size grows, *Cognac* effectively retains its advantage, demonstrating its scalability and robustness.

**Overall**, our work makes progress on the problem of corrective unlearning in graph neural networks with remarkably minimal training signal: we achieve strong unlearning with the knowledge of as little as (5%) of the manipulation set $S_m$. The visualization of the hidden GNN layer embeddings after unlearning (Figure 5 (b)) shows that *Cognac*, like the Oracle model, achieves clear clustering of the manipulated class data points, validating our theoretical results shown in Section 3.1.

# 6. Related Work

Graph-based attacks, such as Sybil (Douceur, 2002) and link spam farms (Wu & Davison, 2005), have long affected the integrity of social networks and search engines. Recent works reveal that even state-of-the-art GNN architectures are vulnerable to simple attacks on the trained models, which either manipulate existing edges and nodes or inject new adversarial nodes (Sun et al., 2019; Dai et al., 2018; Zügner & Günnemann, 2019; Geisler et al., 2024). Parallelly, works have characterized the influence of specific nodes and edges that can guide such attacks (Chen et al., 2023). One strategy to mitigate the influence of such attacks is robust pretraining, such as using adversarial training (Yuan et al., 2024; Zhang et al., 2023). Post-hoc interventions like unlearning act as a complementary layer of defense, helping model developers when attacks slip through and affect a trained model.

Removing the impact of manipulated entities begins with their identification (Brodley & Friedl, 1999), for which multiple strategies exist like data attribution (Ilyas et al.,

*Table 3.* **Ablating both components of *Cognac* across datasets.** CoGN alone does not improve $Acc_{aff}$ over the original model. Conversely, $AC\natural DC$ alone achieves better performance than CoGN, but is still far from matching the Oracle. These results show how both components are integral to *Cognac*'s success.

| METHOD | CS | | CORA | |
|---|---|---|---|---|
| | $Acc_{aff}$ | $Acc_{rem}$ | $Acc_{aff}$ | $Acc_{rem}$ |
| ORIGINAL | 44.3 | 89.8 | 51.4 | 60.6 |
| ORACLE | 90.1 | 90.0 | 71.1 | 60.7 |
| CoGN | 47.2 | **89.8** | 42.1 | 60.5 |
| $AC\natural DC$ | 67.2 | 86.6 | 59.9 | **61.3** |
| *Cognac* | **79.3** | 82.3 | **75.5** | 56.6 |

2022), adversarial detection, and automated or human-in-the-loop anomaly detection (Northcutt et al., 2021). While approaches like model debiasing and concept erasure (Fan et al., 2024; Belrose et al., 2023) can remove effects of identified manipulations post-training, they require knowledge of the manipulation. In contrast, unlearning methods are particularly valuable in adversarial settings where corruption effects may be deliberately obfuscated and impact multiple model behaviors simultaneously (Paleka & Sanyal, 2023).

Recently, machine unlearning has received newfound attention beyond privacy applications (Pawelczyk et al., 2024; Schoepf et al., 2024; Li et al., 2024a;b). Goel et al. (2024) demonstrated the distinction between the *Corrective* and *Privacy-oriented* unlearning settings for *i.i.d.* classification tasks, emphasizing challenges when not all manipulated data is identified for unlearning.

*Exact Unlearning* arose in privacy applications, offering guaranteed removal of data influence through selective retraining (Chen et al., 2022b;a; Bourtoule et al., 2021). While perfect guarantees are valuable for privacy, the exponential cost of sequential deletions (Warnecke et al., 2023) becomes impractical as unlearning expands to broader challenges like model correction and debiasing (Pawelczyk et al., 2024; Schoepf et al., 2024; Li et al., 2024a). This has driven the development of *Inexact Unlearning* methods that balance effectiveness with scalability, using either theoretical bounds for simple models (Chien et al., 2022; Wu et al., 2023b) or empirical validation for deep networks (Wu et al., 2023a; Cheng et al., 2023; Li et al., 2024c). The recent formalization of *Corrective Unlearning* (Goel et al., 2024) opened a crucial new direction: removing corruption effects with only partial identification of manipulated data. We tackle this challenge in graphs, where the non-*i.i.d.* nature of data (Said et al., 2023) results in the graph elements exerting a strong influence on other elements in their neighborhood.

## 7. Limitations and Conclusion

Our work addresses corrective unlearning for GNNs, focusing on removing the effects of manipulated training data when only a fraction is identified. As our method relies on the homophily assumption and is evaluated primarily on homophilic datasets like previous works (Li et al., 2024c), it is left for future works to expand this to heterophilic datasets. While our method does not guarantee successful unlearning against arbitrary real-world attacks, our experiments suffice to show that existing unlearning methods struggle even with complete knowledge of manipulations, which is an unrealistic scenario.

*Cognac* pushes the frontiers of unlearning by consistently beating retraining-from-scratch, and nearly matching the performance of a strong oracle on unlearning class con-

fusion manipulations with access to as little as 5% of the manipulated data. We hope this sparks interesting future work on developing stronger evaluations and theoretical understanding for graph corrective unlearning in the GNN Robustness and Machine Unlearning community.

## Acknowledgements

Varshita Kolipaka is grateful to be funded by IHUB-Data. Arvindh Arun was funded by the CHIPS Joint Undertaking (JU) under grant agreement No. 101140087 (SMARTY), and by the German Federal Ministry of Education and Research (BMBF) under the sub-project with the funding number 16MEE0444, and acknowledges support from the International Max Planck Research School for Intelligent Systems (IMPRS-IS) and the European Laboratory for Learning and Intelligent Systems (ELLIS) PhD programs.

The authors extend their appreciation to the members of the Precog research group, Shashwat Singh, Karuna Chandra, Pratyaksh Gautam, Prashant Kodali, and Makarand Tapaswi for helpful discussions and feedback.

## Impact Statement

The increasing adoption of Graph Neural Networks (GNNs) in real-world applications raises concerns about fairness and safety, particularly in settings where biased data propagates through message passing or incorrect information influences high-stakes decisions. From a fairness perspective, *Cognac* can be potentially used to mitigate the impact of biased social connections in recommender systems and hiring networks, where unfair edges or annotations may reinforce discriminatory patterns. From a safety standpoint, GNNs are increasingly used in drug discovery and biomedical applications, where incorrect relationships between molecular compounds or erroneous biological interactions could lead to misleading predictions. Corrective unlearning can help remove the influence of faulty training data or adversarially inserted edges, improving the reliability of GNN-based scientific models without compromising efficiency.

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

# Appendix

# A. Proof of Theoretical Results

## A.1. Proof of Lemma 3.2

**Lemma.** *Let $G = (V, E)$ be an undirected graph, and let $\mathcal{N}(v)$ denote the 1-hop neighborhood of node $v$. For a node $s \in V$, let $\mathcal{N}^n(s)$ denote the $n$-hop neighborhood of $s$, defined recursively as:*

$$\mathcal{N}^n(s) = \begin{cases} \{s\} & \text{if } n = 0, \\ \bigcup_{v \in \mathcal{N}^{n-1}(s)} \mathcal{N}(v) & \text{if } n > 0 \end{cases}$$

*In an $n$-layer GNN, the representation $z_s$ of node $s$ can affect the representations $z_v$ of nodes $v$ only if $v \in \mathcal{N}^n(s)$. For any $v \notin \mathcal{N}^n(s)$, $z_v$ is independent of $z_s$.*

*Proof.* We prove the claim by induction on the number of layers $l$ in the GNN.

**Base Case ($l = 0$):**
At layer $l = 0$, the representation of a node $v$, $h_v^{(0)}$, is initialized based only on the node's features. Thus, $z_s = h_s^{(0)}$ can only affect $h_s^{(0)}$ itself, and no other node $v \neq s$ is affected.

**Inductive Hypothesis:**
Assume that at layer $l$, the representation $z_s = h_s^{(l)}$ affects only the representations $h_v^{(l)}$ of nodes $v \in \mathcal{N}^l(s)$, the $l$-hop neighborhood of $s$.

**Inductive Step:**
At layer $l + 1$, the representation $h_v^{(l+1)}$ of a node $v$ is computed as:

$$h_v^{(l+1)} = \phi\left(h_v^{(l)}, \psi\left(\{h_u^{(l)} : u \in \mathcal{N}(v)\}\right)\right)$$

By the inductive hypothesis, $h_u^{(l)}$ depends only on nodes in $\mathcal{N}^l(u)$. Therefore, $h_v^{(l+1)}$ depends on nodes in:

$$\bigcup_{u \in \mathcal{N}(v)} \mathcal{N}^l(u)$$

Since $\mathcal{N}^{l+1}(v) = \bigcup_{u \in \mathcal{N}(v)} \mathcal{N}^l(u)$, $h_v^{(l+1)}$ depends only on nodes in $\mathcal{N}^{l+1}(v)$.

For $v$ to be influenced by $z_s$, there must exist a path of length at most $l + 1$ from $s$ to $v$, i.e., $v \in \mathcal{N}^{l+1}(s)$.

**Conclusion:**
By induction, after $n$ layers, $z_s$ can only affect nodes in $\mathcal{N}^n(s)$. For any $v \notin \mathcal{N}^n(s)$, the representation $z_v$ is independent of $z_s$. $\square$

## A.2. Proof of Theorem 3.3

**Theorem.** *Let $\theta \in \mathbb{R}^d$ parameterize a GNN generating embeddings $z_v, z_p, z_n \in \mathbb{R}^k$ for triplets $(v, p, n)$. Let $\mathcal{L}(\theta) = -\mathbb{E}\left[\log \sigma(z_v^\top z_p) + \log \sigma(-z_v^\top z_n)\right]$ and $S(\theta) = \mathbb{E}\left[z_v^\top z_p - z_v^\top z_n\right]$.*

*Then under the assumptions stated below, gradient descent on $\mathcal{L}(\theta)$ with step size $\eta \leq \frac{1}{L}$ (where $L$ is the Lipschitz constant of $\nabla_\theta \mathcal{L}$) guarantees:*

$$S(\theta_{t+1}) \geq S(\theta_t) \quad \forall t \geq 0,$$

*and at convergence, $S(\theta^*) > S(\theta_0)$.*

**Assumption A.1** (Differentiability). *$z_v^\top z_p$ and $z_v^\top z_n$ are differentiable in $\theta$.*

**Assumption A.2** (Convexity). *$\mathcal{L}(\theta)$ is convex in $\theta$.*

**Assumption A.3** (Bounded Gradients). *$\|\nabla_\theta(z_v^\top z_p)\| \leq G$ and $\|\nabla_\theta(z_v^\top z_n)\| \leq G$ for some $G > 0$.*

*Proof.* Let $a = z_v^\top z_p$ and $b = z_v^\top z_n$. Then:

$$\mathcal{L}(\theta) = -\mathbb{E}\left[\log \sigma(a) + \log \sigma(-b)\right]$$

Under Assumption 1 and using $\frac{d}{dx}(\log \sigma(x)) = 1 - \sigma(x)$ and $\frac{d}{dx}(\log \sigma(-x)) = -\sigma(x)$, we compute:

$$\nabla_\theta \mathcal{L} = -\mathbb{E}\left[\underbrace{(1 - \sigma(a))}_{\text{Positive}} \nabla_\theta a - \underbrace{\sigma(b)}_{\text{Positive}} \nabla_\theta b\right]$$

The gradient of $S(\theta)$ is:

$$\nabla_\theta S = \mathbb{E}\left[\nabla_\theta a - \nabla_\theta b\right]$$

The gradient of $\mathcal{L}$ can be rewritten as:

$$\nabla_\theta \mathcal{L} = -\mathbb{E}\left[\sigma(-a)\nabla_\theta a - \sigma(b)\nabla_\theta b\right]$$

Both $\sigma(-a)$ and $\sigma(b)$ are positive for all finite $a, b$. Thus, $\nabla_\theta \mathcal{L}$ is a *negatively weighted combination* of $\nabla_\theta a$ and $\nabla_\theta b$, while $\nabla_\theta S$ is their *unweighted difference*.

Next, we show the monotonic improvement of $S(\theta)$. Consider a gradient descent update:

$$\theta_{t+1} = \theta_t - \eta \nabla_\theta \mathcal{L}$$

The change in $S(\theta)$ is:

$$\Delta S = S(\theta_{t+1}) - S(\theta_t)$$

To simplify the analysis, we'll use a first-order approximation. Specifically, for small updates, we approximate:

$$S(\theta_{t+1}) \approx S(\theta_t) + \langle \nabla_\theta S, \theta_{t+1} - \theta_t \rangle,$$

where $\langle \cdot, \cdot \rangle$ denotes the inner product. Substituting $\theta_{t+1} = \theta_t - \eta \nabla_\theta \mathcal{L}(\theta_t)$, we get:

$$\Delta S \approx -\eta \langle \nabla_\theta S, \nabla_\theta \mathcal{L} \rangle$$

Substituting $\nabla_\theta S$ and $\nabla_\theta \mathcal{L}$:

$$\langle \nabla_\theta S, \nabla_\theta \mathcal{L} \rangle = \mathbb{E}[(\nabla_\theta a - \nabla_\theta b)^\top ((1 - \sigma(a))\nabla_\theta a - \sigma(b)\nabla_\theta b)]$$

$$\Delta S \approx \eta \mathbb{E}[\sigma(-a)\|\nabla_\theta a\|^2 + \sigma(b)\|\nabla_\theta b\|^2 - \sigma(-a)\nabla_\theta b^\top \nabla_\theta a - \sigma(b)\nabla_\theta a^\top \nabla_\theta b]$$

But since the dot product is commutative for real-valued column vectors, i.e., $\nabla_\theta b^\top \nabla_\theta a = \nabla_\theta a^\top \nabla_\theta b$,

$$\Delta S \approx \eta \mathbb{E}[\sigma(-a)\|\nabla_\theta a\|^2 + \sigma(b)\|\nabla_\theta b\|^2 - (\sigma(-a) + \sigma(b))(\nabla_\theta a^\top \nabla_\theta b)]$$

This expression contains both positive and negative terms. However, we now use the fact that the sigmoid function $\sigma(x)$ satisfies $0 < \sigma(x) < 1$ for all real values of $x$, which means:

$$((1 - \sigma(a)) = \sigma(-a) > 0) \text{ for } (a \in \mathbb{R}) \text{ and } (\sigma(b) > 0) \text{ for } (b \in \mathbb{R})$$

Thus, both terms $\sigma(-a)\|\nabla_\theta a\|^2$ and $\sigma(b)\|\nabla_\theta b\|^2$ are positive. On the other hand, the cross-product terms $\nabla_\theta a^\top \nabla_\theta b$ might be negative, but their magnitude is bounded by the gradients $\|\nabla_\theta a\|$ and $\|\nabla_\theta b\|$, which are constrained by the assumption of bounded gradients (Assumption 3). Under this assumption, the gradients $\|\nabla_\theta a\|$ and $\|\nabla_\theta b\|$ are bounded by $G$, i.e., $\|\nabla_\theta a\| \leq G$ and $\|\nabla_\theta b\| \leq G$. Therefore, the cross-product terms are also bounded:

$$|\nabla_\theta a^\top \nabla_\theta b| \leq G^2$$

Now we can write the change in $S(\theta)$ as:

$$\Delta S \approx \eta \mathbb{E}\left[\sigma(-a)\|\nabla_\theta a\|^2 + \sigma(b)\|\nabla_\theta b\|^2 - 2G^2\right]$$

For sufficiently small $\eta$, the positive terms $\sigma(-a)\|\nabla_\theta a\|^2$ and $\sigma(b)\|\nabla_\theta b\|^2$ will dominate the cross-product terms, ensuring that $\Delta S \geq 0$.

$$\Delta S \geq \eta \mathbb{E}\left[\sigma(-a)\|\nabla_\theta a\|^2 + \sigma(b)\|\nabla_\theta b\|^2 - 2G^2\right]$$

For sufficiently small $\eta$, $\Delta S \geq 0$. Thus, $S(\theta_{t+1}) \geq S(\theta_t)$.

By convexity (Assumption 2), gradient descent converges to a global minimum $\theta^*$ where $\nabla_\theta \mathcal{L}(\theta^*) = 0$. At $\theta^*$:

$$\mathbb{E}\left[\sigma(-a)\nabla_\theta a\right] = \mathbb{E}\left[\sigma(b)\nabla_\theta b\right]$$

This equality holds *only* if $\sigma(-a) \to 0$ and $\sigma(b) \to 0$, which occurs when $a \to \infty$ and $b \to -\infty$. Thus:

$$S(\theta^*) = \mathbb{E}[z_v^\top z_p - z_v^\top z_n] \geq S(\theta_0)$$

Strict inequality $S(\theta^*) > S(\theta_0)$ follows from the monotonic improvement at each step.

$\square$

### A.3. Proof of Theorem 3.1

**Theorem.** *Let $G = (V, E, X)$ be a graph with node set $V$, edge set $E$, and features $X$. Let $C_1, C_2 \subset V$ be two distinct classes with ground-truth labels $y_i \in \{C_1, C_2\}$. Suppose an Interclass Confusion (IC) attack is applied as follows:*

1. *Select subsets $S'C_1 \subset C_1$ and $S'C_2 \subset C_2$, each of size $\frac{n}{2}$, forming $S' = S'C_1 \cup S'C_2$*

2. *Swap labels of $\alpha = $ fraction of $S'$, creating a confused set $S_f$*

3. *Train a GNN model $M$ on $(S \setminus S') \cup S_f$*

*Let $\phi_M(C_1), \phi_M(C_2) \in \mathbb{R}^d$ denote the mean embeddings of $C_1$ and $C_2$, and $\mathcal{D}(\phi_M(C_1), \phi_M(C_2))$ be the Wasserstein-2 distance between their embedding distributions. Assume:*

**Assumption A.1** (Homophily preserving GNNs). The GNN uses $L$-layers of homophily-preserving message passing (e.g., mean aggregation).

**Assumption A.2** (Homophily). The graph exhibits $\eta$-homophily, where intra-class edge density dominates inter-class.

**Assumption A.3** (Cross-Entropy Loss). The loss function $\mathcal{L}$ includes cross-entropy and graph smoothness regularization.

*Then, there exists a degradation term $\Delta > 0$ such that:*

$$\mathbb{E}\left[\mathcal{D}(\phi_M(C_1), \phi_M(C_2))\right] \leq \mathbb{E}\left[\mathcal{D}(\phi_{M_{clean}}(C_1), \phi_{M_{clean}}(C_2))\right] - \Delta,$$

*where $\Delta \propto \alpha \cdot \eta \cdot \left(1 - \frac{1}{L}\right)$.*

*Proof.* **Step 1: Embedding Process Formalization**

The GNN computes node embeddings via $L$-layer message passing. For node $v$, the embedding $\phi^{(l)}(v)$ at layer $l$ is:

$$\phi^{(l)}(v) = \sigma\left(\mathbf{W}^{(l)} \cdot \text{AGG}\left(\{\phi^{(l-1)}(u) \mid u \in \mathcal{N}(v)\}\right)\right),$$

where AGG is a mean aggregator, $\mathbf{W}^{(l)}$ are learnable weights, and $\sigma$ is a nonlinearity. The final embedding $\phi_M(v) = \phi^{(L)}(v)$.

**Step 2: Loss Function and Label Noise**

The training loss $\mathcal{L} = \mathcal{L}_{\text{CE}} + \lambda \mathcal{L}_{\text{reg}}$, where:

- $\mathcal{L}_{\text{CE}} = -\frac{1}{|S|} \sum_{v \in S} y_v \log \hat{y}_v$ (cross-entropy)

- $\mathcal{L}_{\text{reg}} = \frac{1}{|E|} \sum_{(u,v) \in E} \|\phi_M(u) - \phi_M(v)\|^2$ (smoothness regularization)

For $S_f$, labels $y_v$ are swapped between $C_1$ and $C_2$. Let $\mathcal{E}_{\text{noise}} = \{v \in S_f \mid y_v \text{ is incorrect}\}$. The corrupted $\mathcal{L}_{\text{CE}}$ forces conflicting gradients for $v \in \mathcal{E}_{\text{noise}}$, pulling $\phi_M(v)$ toward the wrong class centroid.

**Step 3: Bias in Mean Embeddings**

Let $\mu_{C_1}, \mu_{C_2}$ be the mean embeddings of $C_1, C_2$ under $M_{\text{clean}}$. After IC attack, for $v \in \mathcal{E}_{\text{noise}}$:

$$\mathbb{E}[\phi_M(v)] = \alpha \mu_{C_2} + (1 - \alpha)\mu_{C_1} \quad (\text{if } v \in C_1)$$

By linearity of expectation, the perturbed mean for $C_1$ becomes:

$$\mu'_{C_1} = \mu_{C_1} - \alpha(\mu_{C_1} - \mu_{C_2})$$

Similarly, $\mu'_{C_2} = \mu_{C_2} + \alpha(\mu_{C_1} - \mu_{C_2})$. Thus, the distance between means reduces by $2\alpha\|\mu_{C_1} - \mu_{C_2}\|$.

**Step 4: Message-Passing Amplification**

Under $\eta$-homophily (where $\eta$ is the 'extent' of homophily), neighbors of $v \in \mathcal{E}_{\text{noise}}$ are likely in $C_1$. The smoothness term $\mathcal{L}_{\text{reg}}$ propagates the corrupted embedding of $v$ to its neighbors, perturbing their embeddings. After $L$ layers, the influence of $\mathcal{E}_{\text{noise}}$ spreads to $\sim \eta^L|V|$ nodes. This amplifies the mean embedding shift by a factor $\eta\left(1 - \frac{1}{L}\right)$.

**Step 5: Wasserstein Distance Bound**

The Wasserstein-2 distance between $\phi_M(C_1)$ and $\phi_M(C_2)$ is dominated by the mean shift and covariance distortion. Using the Fréchet inequality:

$$\mathcal{D}^2 \leq \|\mu'_{C_1} - \mu'_{C_2}\|^2 + \text{Tr}(\Sigma_{C_1} + \Sigma_{C_2} - 2(\Sigma_{C_1}\Sigma_{C_2})^{1/2})$$

From Steps 3 and 4, $\|\mu'_{C_1} - \mu'_{C_2}\|^2 = (1 - 2\alpha\eta(1 - \frac{1}{L}))\|\mu_{C_1} - \mu_{C_2}\|^2$. Thus,

$$\mathbb{E}[\mathcal{D}(\phi_M(C_1), \phi_M(C_2))] \leq \mathbb{E}[\mathcal{D}(\phi_{M_{\text{clean}}}(C_1), \phi_{M_{\text{clean}}}(C_2))] - \Delta,$$

where $\Delta = \alpha\eta\left(1 - \frac{1}{L}\right)\|\mu_{C_1} - \mu_{C_2}\|^2$.

**Conclusion**

The IC attack reduces the separability of $C_1$ and $C_2$ in the embedding space by biasing their mean embeddings toward each other and amplifying this bias via graph convolutions. The degradation $\Delta$ scales with $\alpha, \eta$, and the depth $L$, confirming representation entanglement. □

## B. Data Manipulation Statistics

Table 4 presents the manipulation statistics for all datasets used in our study. We introduce a significant variation in the degree of manipulation to achieve two key objectives: (1) in some datasets, a lower percentage of manipulated nodes or edges is sufficient to induce noticeable performance degradation, while others require more extensive modifications, and (2) we aim to evaluate unlearning performance across a broad spectrum of manipulation scales. For instance, PubMed undergoes the highest level of manipulation, with nearly 39% of training nodes affected and 33.84% additional edges introduced, whereas datasets like CS and CoraFull experience minimal modifications. Additionally, for OGB-Arxiv, a significantly larger dataset, only 3.14% of training nodes are manipulated. This diverse manipulation strategy ensures a rigorous and comprehensive evaluation of unlearning effectiveness across different graph structures.

## C. Formal Description of *Cognac*

In this section, we outline the procedure of our proposed unlearning method, *Cognac*, designed to effectively remove the influence of manipulated data from GNNs. First, the algorithm identifies the nodes affected by the manipulation, as well as

*Table 4.* **Dataset manipulation statistics**. Statistics of manipulations across datasets. The percentage of nodes and edges added are relative to the existing number of training nodes and the number of edges in the graph, respectively.

| DATASET | TRAINING NODES | NODES MANIPULATED (%) | EDGES ADDED (%) |
|---|---|---|---|
| CORAFULL | $11,280$ | 1.45 | 1.20 |
| CORA | $1,491$ | 14.88 | 17.26 |
| CITESEER | $1,272$ | 14.78 | 20.38 |
| DBLP | $9,714$ | 10.50 | 5.78 |
| PUBMED | $11,830$ | 38.96 | 33.84 |
| OGB-ARXIV | $90,941$ | 3.14 | - |
| CS | $10,999$ | 2.25 | 1.83 |
| PHYSICS | $20,695$ | 8.17 | 5.04 |
| PHOTOS | $4,492$ | 11.40 | 5.04 |
| COMPUTERS | $8,028$ | 3.14 | 0.814 |

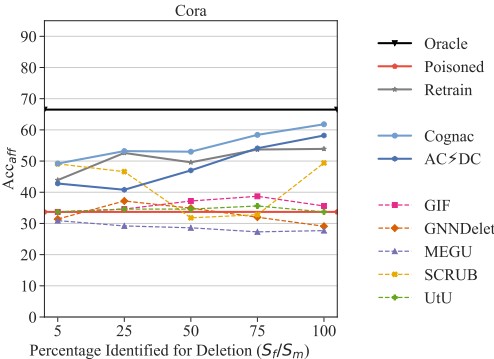

*Figure 6.* **Node Unlearning results for Cora with GAT backbone.** Comparison of $\text{Acc}_{\text{aff}}$ for the unlearning methods on GAT trained on Cora for different values of $(S_f/S_m)$. We see that *Cognac* outperforms all baselines.

their corresponding positive and negative samples, and then alternates between applying CoGN and AC$\frac{1}{2}$DC to unlearn their influence. The key steps include identifying the affected nodes, performing contrastive learning to re-optimize the embeddings, and minimizing classification loss on the unaffected nodes while maximizing it on the discovered manipulated set ($S_f$). The complete algorithm is detailed in Algorithm 1.

## D. Results Showing The Breadth of Applicability of *Cognac*

### D.1. Results on GAT

To provide a comprehensive comparison between *Cognac* and other methods, we provide results on commonly used GNN backbone architectures - GCN and GAT.

**Graph Convolutional Network (GCN)** is a method for semi-supervised classification of graph-structured data. It employs an efficient layer-wise propagation rule derived from a first-order approximation of spectral convolutions on graphs.

**Graph Attention Network (GAT)** employs computationally efficient masked self-attention layers that assign varying importance to neighborhood nodes without needing the complete graph structure upfront, thereby overcoming many theoretical limitations of earlier spectral-based methods.

Figure 6 shows that *Cognac* also performs competitively with a GAT backbone. When 5% of $S_m$ is known, SCRUB performs similarly to *Cognac*, within the standard deviation. For higher fractions, we achieve greater $\text{Acc}_{\text{aff}}$ than the benchmark graph unlearning methods with large margins, often beating the performance of retraining the GNN from scratch. These results indicate that benchmark graph unlearning methods used for comparison cannot recover from the impact of the label flip poison. In contrast, our method is much closer to Oracle's performance.

---

**Algorithm 1** COGNAC

---

**Require:** GNN $M$, Graph $G = (V, E, X)$, Deletion set $S_f$, Hyperparameters $\Theta$
**Ensure:** Unlearned GNN $M^*$
1: $S \leftarrow$ IDENTIFYAFFECTEDNODES$(M, X, S_f, E, \Theta)$
2: $P \leftarrow$ SAMPLEPOSITIVES$(S, E, S_f)$
3: $N \leftarrow$ SAMPLENEGATIVES$(S, S_f)$
4: // Overall unlearning process
5: **for** outer_epoch = 1 to $\Theta_{\text{total\_epochs}}$ **do**
6:     // Contrastive unlearning phase
7:     **for** epoch = 1 to $\Theta_{\text{contrast\_epochs}}$ **do**
8:         $Z \leftarrow M(X)$
9:         $\mathcal{L}_c \leftarrow \sum_{v \in S}(-\log(\sigma(Z_v^T Z_P)) - \log(\sigma(-Z_v^T Z_N)))$
10:        $M \leftarrow$ OPTIMIZE$(M, \mathcal{L}_c)$
11:     **end for**
12:     // Gradient ascent on $S_f$, and gradient descent on $V \setminus S_f$
13:     **for** epoch = 1 to $\Theta_{\text{ascent\_descent\_epochs}}$ **do**
14:         $\mathcal{L}_a \leftarrow -$CROSSENTROPY$(M(X)_{S_f}, Y_{S_f})$
15:         $M \leftarrow$ OPTIMIZE$(M, \mathcal{L}_a)$
16:         $\mathcal{L}_d \leftarrow$ CROSSENTROPY$(M(X)_{V \setminus S_f}, Y_{V \setminus S_f})$
17:         $M \leftarrow$ OPTIMIZE$(M, \mathcal{L}_d)$
18:     **end for**
19: **end for**
20: **return** $M$
21:
22: **function** IdentifyAffectedNodes$(M, X, S_f, E, \Theta)$
23:     $X' \leftarrow$ INVERTFEATURES$(X, S_f, E)$
24:     $\Delta \leftarrow |M(X') - M(X)|$
25:     **return** TOPK$(\Delta, \Theta_k)$
26: **end function**

---

## D.2. Additional Experiments on Large $S_f$

To stress-test *Cognac*'s performance - as methods could potentially degrade as the size of $S_f$ grows - we conduct an experiment where we choose a significant fraction of the training nodes of the Amazon, DBLP, and Physics datasets, to be attacked (by the binary label flip attack) and marked for deletion. The method performs competitively even at this large deletion size. Table 5 demonstrates these results.

*Table 5.* **Evaluation on larger sizes of** $S_f$**.** *Cognac* performs well across datasets (Amazon, DBLP, Physics), even when a significant fraction of the total training nodes are present in $S_f$.

| METHOD | AMAZON (25%) | | DBLP (29.4%) | | PHYSICS (14.8%) | |
|---|---|---|---|---|---|---|
| | $\text{Acc}_{\text{rem}}$ | $\text{Acc}_{\text{aff}}$ | $\text{Acc}_{\text{rem}}$ | $\text{Acc}_{\text{aff}}$ | $\text{Acc}_{\text{rem}}$ | $\text{Acc}_{\text{aff}}$ |
| ORACLE | 92.9 | 95.6 | 72.9 | 86.0 | 95.2 | 95.1 |
| ORIGINAL | 92.5 | 49.0 | 74.9 | 57.9 | 58.9 | 95.4 |
| *Cognac* | **92.4** | **83.7** | **82.3** | 81.7 | **90.7** | **95.0** |
| GNNDELETE | 27.7 | 49.7 | 45.0 | 49.6 | 1.4 | 37.3 |
| SCRUB | 92.3 | 72.6 | 77.5 | **82.1** | 77.9 | 94.9 |

## D.3. Results on a trigger poisoning attack

We expand the analysis to a wider range of attack scenarios, now covering all major poisoning types (label, graph structure, and feature). Our feature attack injects trigger a pattern into the feature vectors of select nodes, and assigning a fixed spurious label, and reduces accuracy on the target distribution. Despite not being the strongest possible attack, our implementation provides sufficient signal for evaluation - most unlearning methods struggle, while Cognac matches and even outperforms retraining performance. Complete results are provided in Table 6.

The attacker selects a subset of victim nodes $S_p \subset V$. For each node $v \in S_p$, its original feature vector $x_v$ is modified to $x'_v$ by setting specific trigger feature indices $j \in I_t$ to 1 (i.e., $x'_{v,j} = 1 \ \forall j \in I_t$, while other features $x_{v,j}$ for $j \notin I_t$ remain unchanged). Subsequently, all nodes $v \in S_p$ are assigned a fixed target label $y_{target}$. The choice of victim nodes here is random within the victim class, and the number is chosen to maintain stealth while maximising Attack Success Rate (the percentage of manipulated nodes in the test set that are successfully classified as the target class).

This use of a localized feature pattern as a trigger is analogous to patch-based backdoor attacks in image poisoning, such as those introduced by BadNets (Gu et al., 2017).

*Table 6.* **Results for unlearning the trigger-based feature manipulation.** *Cognac* performs strongly across datasets, maintaining a high $\text{Acc}_{\text{aff}}$ and $\text{Acc}_{\text{rem}}$. Retrain fails to recover accuracy on the victim class for the *Cora* dataset, while SCRUB fails to do so for both *Cora* and *CS*. GNNDelete and MEGU, the graph unlearning baselines, fail to remove the poison.

| METHOD | PHOTOS | | CORA | | CS | |
|---|---|---|---|---|---|---|
| | $\text{Acc}_{\text{rem}}$ | $\text{Acc}_{\text{aff}}$ | $\text{Acc}_{\text{rem}}$ | $\text{Acc}_{\text{aff}}$ | $\text{Acc}_{\text{rem}}$ | $\text{Acc}_{\text{aff}}$ |
| ORIGINAL | $95.0 \pm 0.0$ | $33.9 \pm 0.0$ | $67.4 \pm 0.0$ | $25.6 \pm 0.0$ | $89.3 \pm 0.0$ | $0.0 \pm 0.0$ |
| RETRAIN | $94.1 \pm 0.5$ | $92.0 \pm 1.7$ | $\mathbf{68.3 \pm 0.4}$ | $46.9 \pm 6.8$ | $93.0 \pm 0.2$ | $92.8 \pm 2.3$ |
| GNNDELETE | $17.8 \pm 0.0$ | $0.0 \pm 0.0$ | $65.7 \pm 0.2$ | $40.8 \pm 9.1$ | $68.0 \pm 0.0$ | $0.0 \pm 0.0$ |
| MEGU | $81.5 \pm 8.7$ | $17.5 \pm 23.1$ | $61.5 \pm 0.3$ | $1.0 \pm 1.0$ | $88.7 \pm 0.1$ | $0.0 \pm 0.0$ |
| SCRUB | $93.8 \pm 0.0$ | $\mathbf{92.7 \pm 0.0}$ | $68.0 \pm 0.0$ | $64.1 \pm 0.0$ | $92.7 \pm 0.1$ | $72.1 \pm 21.8$ |
| *Cognac* | $\mathbf{94.6 \pm 0.0}$ | $91.1 \pm 0.0$ | $67.3 \pm 0.0$ | $\mathbf{78.2 \pm 0.0}$ | $\mathbf{93.3 \pm 0.0}$ | $\mathbf{93.6 \pm 0.6}$ |

# E. Convergence and Ablations of *Cognac*

## E.1. Convergence

We now discuss the convergence properties of *Cognac*. Plots in Figure 7 describe the losses of each of the components of our method (contrastive, ascent, descent) after the last epoch of every *step*, the meaning of which should be clear from Algorithm 1: Line 4 (which we denote as `num_steps`). The loss plots are constructed over the best hyperparameters, and

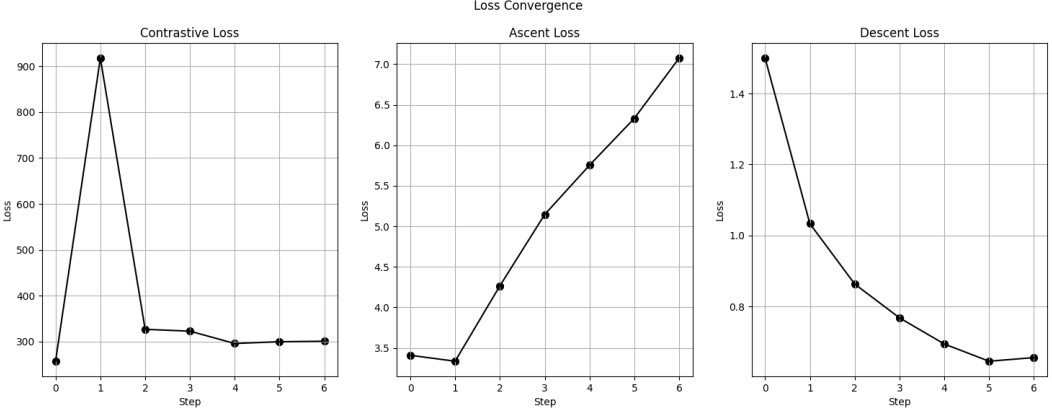

*Figure 7.* **Convergence of the losses across unlearning steps**. The ascent loss on $S_m$ continually increases as expected, the descent loss on $S \setminus S_m$ converges, and the contrastive loss exhibits a low plateau after an initial overshoot, implying it may have learned discriminative features.

we would likely not see such convergence trends with sub-optimal hyperparameters, which may provide insights to improve performance when it's used in other settings as well.

### E.2. Analysis of method used to find affected neighbors

Our strategy to find affected neighbors is likely not perfect for finding the most affected nodes, and more sophisticated influence functions, such as the one presented in (Chen et al., 2023), could be used to potentially improve performance. Still, we note that it achieves a $5\%$ higher $\mathrm{Acc_{aff}}$ than while choosing random $k\%$ nodes in the $n$-hop neighborhood (where $n$ is the number of layers of message passing) while being cheap to compute: we only require a single forward pass over the model with the inverted features. Interestingly, Figure 8 also shows that even if the GNN is not well-trained, if we choose the top $k\%$ affected nodes, the unlearning performance does not change much, while still being noticeably better than when we use a random $k\%$ of the neighbors.

Figure 9 (left) shows that there are no noticeable changes in taking a smaller or larger $k\%$. However, removing this step entirely ($k = 0\%$) results in worse performance, suggesting that performing contrastive unlearning on even a small $k\%$ is significant. Additionally, by keeping this percentage small, we ensure computational efficiency without diminishing performance (Figure 9 (right)), which is essential for unlearning methods.

### E.3. Comparing our sampling method to MEGU's

Below, we compare the performance of *Cognac* while using our sampling method (described in Section 3.1.1) against its performance when using MEGU's sampling method.

*Table 7.* **Evaluating *Cognac* with a different strategy to identify affected nodes on *Cora*.** We find that our strategy outperforms the MEGU's, while also being 8x faster. Both variants are hyperparameter-tuned for 100 runs.

| METHOD | $\mathrm{Acc_{rem}}$ | $\mathrm{Acc_{aff}}$ | SAMPLING TIME (3160 NODES) |
|---|---|---|---|
| ORIGINAL | $61.4 \pm 0.00$ | $40.0 \pm 0.00$ | - |
| ORACLE | $58.4 \pm 0.00$ | $69.2 \pm 0.00$ | - |
| *Cognac* (MEGU'S) | $54.8 \pm 0.00$ | $48.7 \pm 0.00$ | $0.432\,s$ |
| *Cognac* (OURS) | $\mathbf{56.6 \pm 0.00}$ | $\mathbf{75.5 \pm 0.04}$ | $\mathbf{0.054\,s}$ |

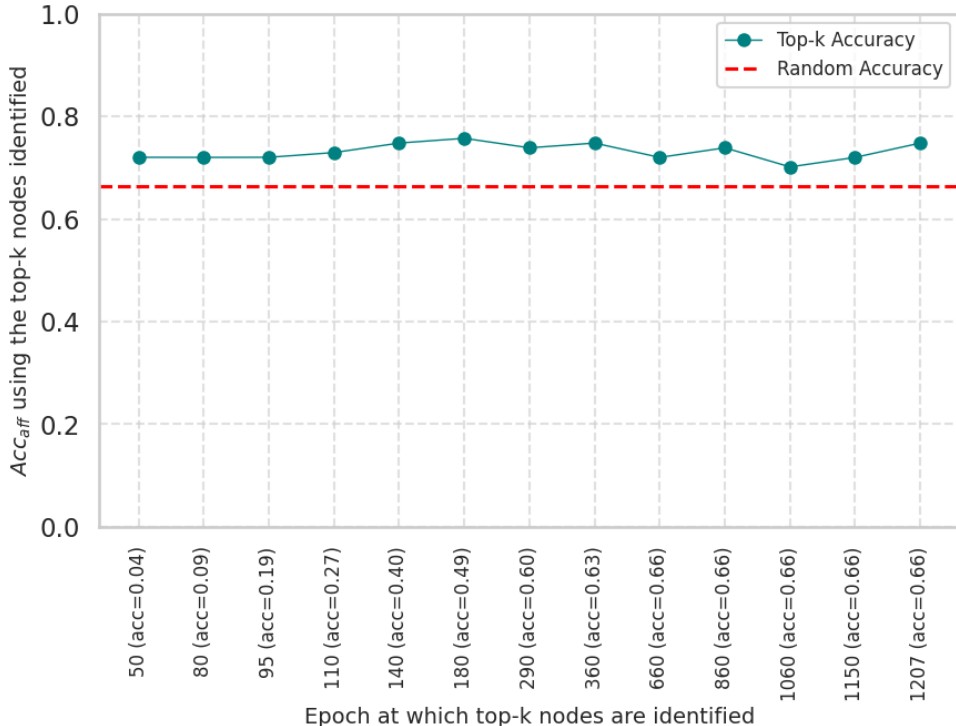

*Figure 8.* **Effect of how well-trained the GNN is on top-$k\%$ affected neighbor identification.** Here $k = 4$. The x-axis represents the epoch at which we used GNN representations to identify the most affected neighbors for *Cognac*. The y-axis reports the unlearning performance after contrastive unlearning on these identified nodes using the final model. The red line contains performance after picking a random subset from the $n$-hop neighborhood. Affected neighborhood identification using top-$k\%$ logit change is more effective even with an extremely undertrained GNN.

### E.4. Why does $\mathrm{Acc_{aff}}$ sometimes reduce as identified manipulated entities increase?

We find an interesting trend that sometimes, as more of the manipulation set $(S_m)$ is known and used as the deletion set $(S_f)$ (going left to right in Figure 4), $\mathrm{Acc_{aff}}$ reduces. This can seem counterintuitive, as one would expect the accuracy of affected classes to improve as more samples are used for unlearning. We hypothesize that unlearning a larger fraction of the manipulation set reduces $\mathrm{Acc_{aff}}$ due to two factors that adversely affect the neighborhoods of the nodes removed, which typically have other nodes of the affected classes due to homophily. First, in the case of label manipulation, when we model it as node unlearning for consistency with prior work, we lose correct information about the graph structure. Second, when modifying the graph structure, i.e., removing some edges or nodes, changes the feature distribution of their neighboring nodes after the message passes, making it out of distribution for the learned GNN layers. The same rationale is why the test nodes are kept in the graph structure (without optimizing the task loss for them) during training (Kipf & Welling, 2017). We investigate this by adding an ablation where, in the unlearning of the label manipulation, instead of unlearning the whole node, we keep the structure, i.e., the node and connected edges, but unlearn the features and labels.

As observed in Table 8, retaining the node structure leads to large improvements in $\mathrm{Acc_{aff}}$ when the deletion set is larger (the full set of manipulated entities), while not benefiting much when the deletion set is smaller. In the full manipulation set deletion setting, *Cognac* even slightly outperforms Oracle. This highlights how, unlike conventional node unlearning in graphs, removing the nodes is not always the best way to unlearn manipulations. They can simply be moved from the train set to the test set to still partake in message passing, so the task loss is not optimized over wrong labels.

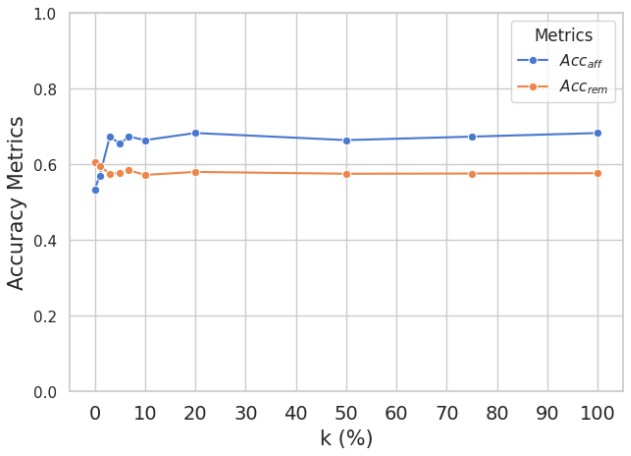 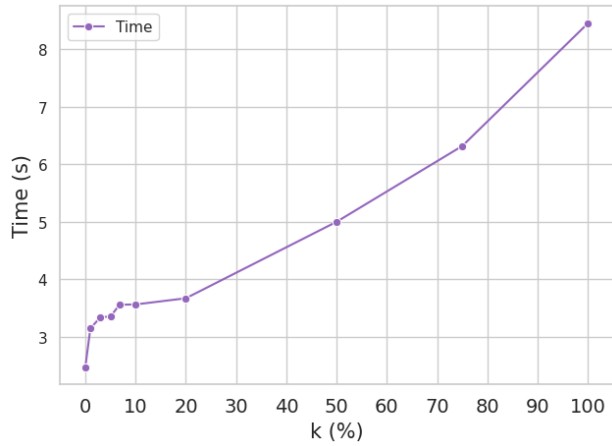

*Figure 9.* **Effect of $k$ on unlearning ($\text{Acc}_{\text{aff}}$), utility ($\text{Acc}_{\text{rem}}$) and efficiency when identifying top-$k$% affected nodes for contrastive unlearning.** (Left) *Cognac* effectiveness sharply improves beyond $k = 0$%, suggesting that performing contrastive unlearning on even a small percentage of nodes ($k$) significantly enhances the algorithm's effectiveness. However, using higher values of $k$ yields similar performance with the added downside of increasing computational time (Right).

*Table 8.* **Ablating node unlearning performance on label manipulation with and without unlinking.** We report the accuracy on the affected classes $\text{Acc}_{\text{aff}}$ for unlearning the label manipulation on Cora, both when the full and a subset (25%) of the manipulated set is used for deletion. We find that not removing the structural information leads to a significant improvement in $\text{Acc}_{\text{aff}}$, especially when more entities are deleted. This illustrates that the unlearning methods can achieve improved performance when precise information about the manipulated data is available.

| Method | 0.25 | | 1.00 | |
|---|---|---|---|---|
| | **Linked** | **Unlinked** | **Linked** | **Unlinked** |
| Oracle | $73.0_{\pm 0.0}$ | $73.0_{\pm 0.0}$ | $73.0_{\pm 0.0}$ | $73.0_{\pm 0.0}$ |
| Original | $42.0_{\pm 0.0}$ | $42.0_{\pm 0.0}$ | $42.0_{\pm 0.0}$ | $42.0_{\pm 0.0}$ |
| *Cognac* | $\mathbf{64.8_{\pm 0.9}}$ | $\mathbf{67.8_{\pm 3.2}}$ | $\mathbf{77.2_{\pm 1.0}}$ | $\mathbf{69.3_{\pm 1.3}}$ |
| GNNDelete | $35.2_{\pm 2.5}$ | $50.2_{\pm 1.9}$ | $21.9_{\pm 4.5}$ | $30.2_{\pm 5.3}$ |
| MEGU | $40.8_{\pm 1.6}$ | $33.4_{\pm 0.4}$ | $41.2_{\pm 1.5}$ | $32.1_{\pm 1.3}$ |
| SCRUB | $45.7_{\pm 0.0}$ | $60.7_{\pm 0.0}$ | $41.1_{\pm 0.0}$ | $29.0_{\pm 0.0}$ |
| AC$\notmid$DC | $61.7_{\pm 0.0}$ | $58.8_{\pm 0.0}$ | $63.7_{\pm 0.0}$ | $54.3_{\pm 0.0}$ |

# F. Detailed report of the experimental setup

## F.1. Hyperparameter Tuning

We perform extensive hyperparameter tuning for all unlearning methods using Optuna (Akiba et al., 2019) with a TPESampler (Tree-structured Parzen Estimator) Algorithm. We ensure the hyperparameter ranges searched include any values specified by the methods. The optimization target is an average of $\text{Acc}_{\text{aff}}$ and $\text{Acc}_{\text{rem}}$, computed on the validation set. We report averaged results across five seeds. Method-specific hyperparameter ranges and scatter plots across hyperparameters for each method are provided in Appendix F.1.

We perform hyperparameter tuning for each combination of attack, dataset, unlearning method, and the identified fraction of deletion set ($S_f$). The optimization target is an average of $\text{Acc}_{\text{aff}}$ and $\text{Acc}_{\text{rem}}$, computed on the validation set. For each setting, we run 100 trials with hyperparameters selected using the TPESampler (Tree-structured Parzen Estimator) algorithm. In Figures 10, 11 and 12, we report $\text{Acc}_{\text{aff}}$ and $\text{Acc}_{\text{rem}}$ scores for each hyperparameter tuning trial. Across hyperparameter runs, existing graph-based unlearning methods, barring MEGU, vary drastically across different sets of hyperparameters. On the other hand, our proposed method *Cognac* and its ablations show consistently high scores across hyperparameters, showcasing *Cognac*'s robustness to hyperparameter tuning.

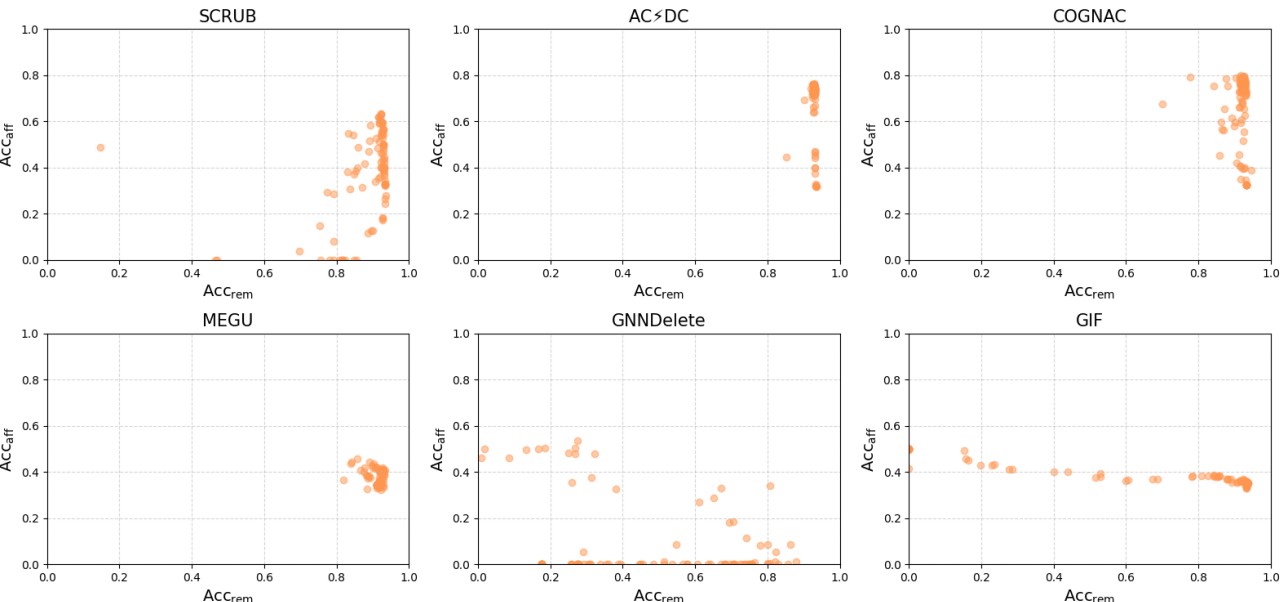

*Figure 10.* **Hyperparameter runs for $S_f = 1.0$ on Amazon**. Scores of various hyperparameter trial runs. The best hyperparameters are selected according to the run achieving the best value for the average of $\mathrm{Acc_{rem}}$ and $\mathrm{Acc_{aff}}$.

## F.2. Unlearning Times

To ensure the practicality of inexact unlearning methods, they must achieve greater efficiency than retraining from scratch. As illustrated in Figure 13, *Cognac* demonstrates competitive efficiency compared to alternative methods, offering substantial speedups over retraining from scratch.

**Measuring Unlearning Time.** To simplify comparisons to just two axes, $\mathrm{Acc_{aff}}$, and $\mathrm{Acc_{rem}}$, we fix a maximum cutoff of time an unlearning method can take, as motivated by Maini et al. (2024). We chose this cutoff as 25% of the original model training time. We pick the best model checkpoint during training for each method, which could be achieved earlier than this. Average run times for each method reported in Figure 13 under Appendix F.2 show that *Cognac*'s efficiency is comparable to or better than baselines. All experiments were run on a machine with Intel Xeon CPUs and two dedicated RTX 5000 GPUs.

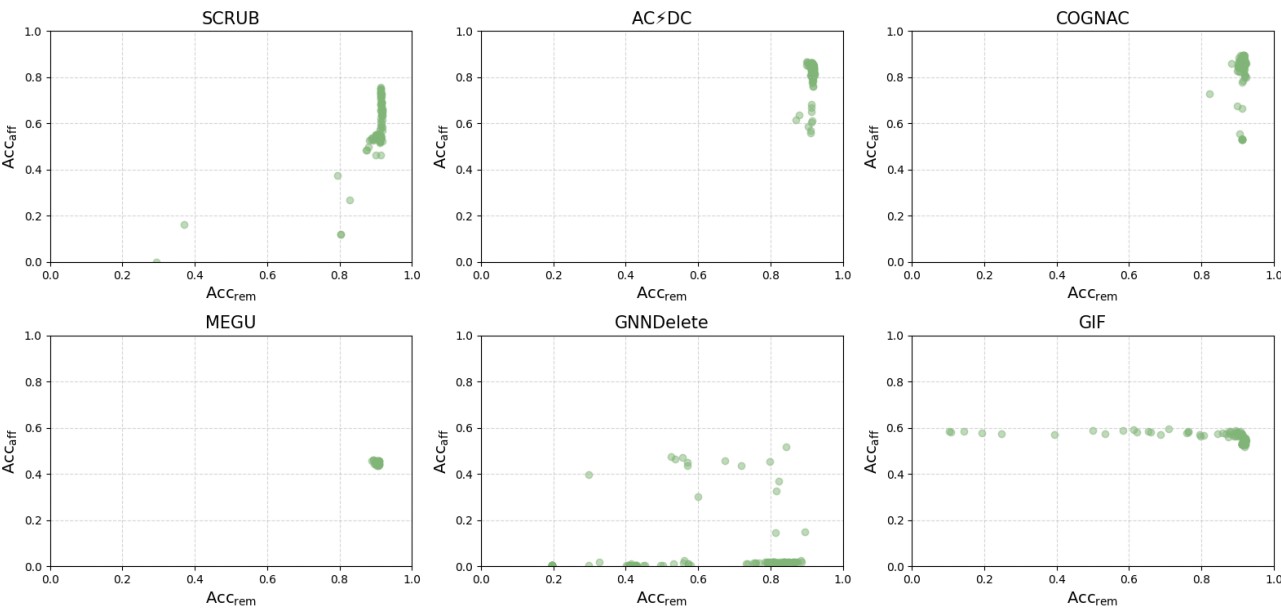

*Figure 11.* **Hyperparameter runs for** $S_f = 1.0$ **on CS**. Scores of various hyperparameter trial runs. The best hyperparameters are selected according to the run achieving the best value for the average of $\mathrm{Acc_{rem}}$ and $\mathrm{Acc_{aff}}$.

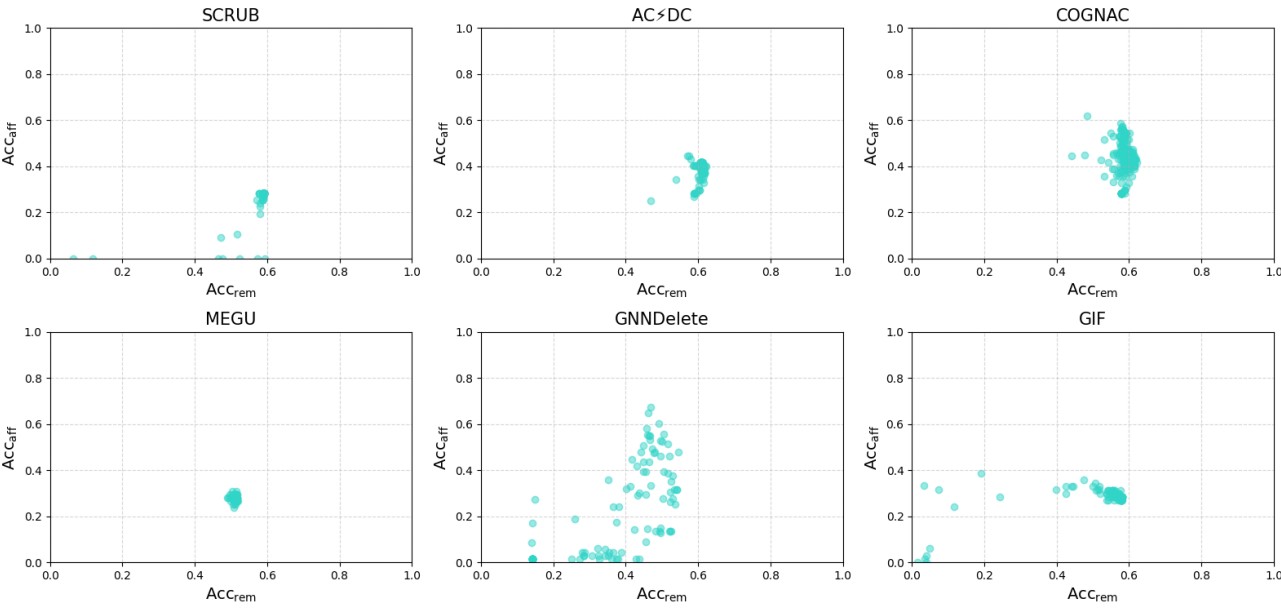

*Figure 12.* **Hyperparameter runs for** $S_f = 1.0$ **on Cora**. Scores of various hyperparameter trial runs. The best hyperparameters are selected according to the run achieving the best value for the average of $\mathrm{Acc_{rem}}$ and $\mathrm{Acc_{aff}}$.

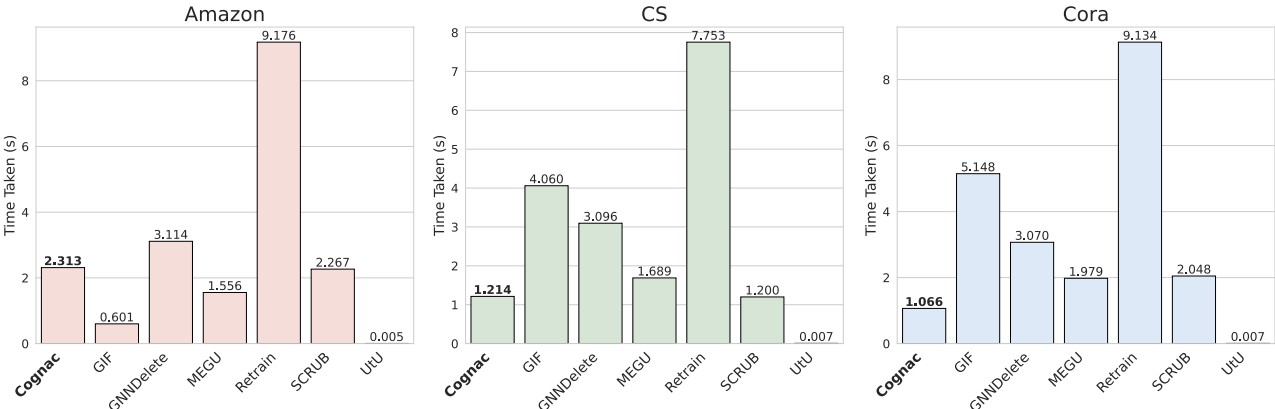

*Figure 13.* **Time taken to unlearn.** We report the time taken by each unlearning method for all datasets, in the setting where all manipulated samples are known for deletion. *Cognac* provides significant speedups over Retrain and has performance similar to other unlearning methods.

