# OpenReview forum: "A Cognac Shot To Forget Bad Memories: Corrective Unlearning for Graph Neural Networks"
_ICML.cc/2025/Conference — ICML 2025 poster_

### Official Review · Reviewer_Lw8o · 2025-03-15

**Overall Recommendation:** 3

**Summary:**

The authors propose a methodology, named Cognac, aimed at enhancing the fairness, robustness, and accuracy of Graph Neural Networks (GNNs) through corrective unlearning techniques applied to specific nodes. Cognac consists of two primary components:

1.	Contrastive Unlearning on Graph Neighborhoods (CoGN): This component identifies nodes influenced by entities that should be removed and strategically repositions them away from those entities, while concurrently moving them closer to normal nodes through contrastive learning.

2.	AsCent DesCend de-coupled (ACDC): This technique leverages gradient ascent and descent methods to induce effective unlearning for sets of nodes designated for removal.

The authors support their proposed components with sufficient mathematical justification and rigorous proofs, demonstrating the robustness and theoretical soundness of the proposed methodology. Extensive experiments conducted across diverse benchmark datasets and multiple GNN models illustrate the superiority of Cognac over existing approaches. Notably, the proposed methodology maintains strong performance even when the number of nodes designated for removal is extremely limited and consistently scales effectively to significantly larger datasets.

**Claims And Evidence:**

The authors provide clear and convincing evidence supporting the two core components of their proposed methodology (CoGN and ACDC). Specifically, they present rigorous mathematical justifications for identifying and adjusting nodes through the CoGN approach, including the derivation of the criteria used to select nodes for contrastive unlearning. Additionally, the authors offer comprehensive theoretical proofs regarding the ACDC technique, which effectively unlearns the influence of designated node sets. Beyond these theoretical validations, the authors also perform extensive experimental evaluations demonstrating that their proposed methods outperform existing state-of-the-art techniques in various experimental scenarios.

**Essential References Not Discussed:**

NA

**Experimental Designs Or Analyses:**

The authors demonstrate the validity of their proposed method by comparing it with four primary GNN unlearning techniques (original model, retrain, finetune, i.id.) and five state-of-the-art benchmark models. They emphasize that the widely adopted retrain baseline, as indicated by recent research, does not represent an optimal standard for corrective unlearning. In their evaluation, the authors convincingly show that their proposed method outperforms the retrain-based approach. Furthermore, the authors introduce an oracle baseline trained on the full, unmodified dataset to represent an upper-bound for corrective unlearning performance. The authors' method demonstrates strong performance relative to both the retrain and oracle baselines, further validating the effectiveness and rigor of their experimental evaluations.

**Methods And Evaluation Criteria:**

The authors evaluate their proposed method using a recently introduced metric from the literature (ICLR, 2024). They clearly indicate the sources of benchmark datasets widely adopted in prior research, ensuring transparency in their evaluations. Moreover, the authors acknowledge that, given the inherent characteristics of unlearning methods, guaranteeing a fair comparison can be challenging. To address this issue, they openly share multiple components essential for reproducibility, effectively demonstrating the robustness of their experimental setup. Finally, the soundness of the experimental approach is further strengthened by thoroughly comparing their methodology against five state-of-the-art benchmark models and four additional reference models.

**Other Comments Or Suggestions:**

NA

**Other Strengths And Weaknesses:**

Strengths
1.	The authors propose a novel approach, named Cognac, comprising two key components—Contrastive Unlearning on Graph Neighborhoods (CoGN) and Ascent Descent De-coupled (ACDC)—to address corrective unlearning in Graph Neural Networks (GNNs). The authors provide rigorous mathematical justifications and proofs supporting both components, demonstrating their theoretical soundness and validity clearly and convincingly.
2.	Through extensive experimental evaluation on various benchmark datasets and models, the authors demonstrate the superiority of Cognac compared to existing baseline unlearning approaches. Notably, the experiments include a wide array of benchmark datasets and state-of-the-art models. The authors also highlight the scalability of their approach by showing its effectiveness on datasets up to eight times larger than standard benchmarks, further supporting the method's robustness.
3.	The authors demonstrate that Cognac achieves superior performance even when compared against the oracle model, which represents the upper bound for corrective unlearning performance. Remarkably, Cognac maintains strong performance despite having access to only 5% of the manipulated data, clearly highlighting the method’s effectiveness and efficiency.

**Questions For Authors:**

NA

**Relation To Broader Scientific Literature:**

The paper extensively compares to prior graph unlearning methods. For instance, GNNDelete by Cheng et al. (2023) is taken as a baseline. By doing so, the authors clarify that they are tackling the corrective post-hoc unlearning scenario rather than the more studied privacy-driven exact unlearning scenario, which is an important distinction.
Furthermore, the paper relates Cognac to general unlearning methods like SCRUB (Kurmanji et al., 2023). By including SCRUB, they acknowledge the broader machine unlearning literature. The fact that Cognac, with graph-specific insight, outperforms SCRUB in this domain highlights the contribution that graph structure awareness brings. In positioning relative to broader literature, the authors also mention robust training and concept erasure approaches in the introduction.
They cite works on robust pre-training for GNNs (adversarial training by Yuan et al., 2024; defense by Zhang et al., 2023) and concept erasure in vision (Belrose et al., 2023) to clarify that while those aim to remove unwanted influences, unlearning is distinct in that it’s post-hoc and does not assume knowledge of the specific concept or attack in advance. This helps readers see how their work is different from training a GNN to be robust to attacks from the start instead fix the model after the fact.

**Theoretical Claims:**

The authors provide rigorous mathematical justification regarding their proposed method. First, they clearly formalize and prove how the Wasserstein-based interclass confusion attack affects class distributions by quantifying changes via mathematical analysis, thereby establishing a strong theoretical foundation. Additionally, the authors offer mathematical proofs to efficiently identify specific cases where the representation of an arbitrary node could be influenced, significantly optimizing the node-selection step in contrastive unlearning (CoGN). Finally, they define a loss function suitable for effective contrastive unlearning and rigorously prove important theoretical properties, including differentiability, convexity, and bounded gradients, thereby further solidifying the soundness and reliability of their approach.

---

> ### Author Rebuttal · Authors · 2025-04-01
>
> We thank the reviewer for their thorough and positive assessment of our work! We are thrilled that you recognized the novelty, mathematical soundness, experimental rigor, and effectiveness of our proposed method. In response to the other reviewers, we have added more experiments: (1) a feature trigger poison (reviewer igz9), (2) ablations of CoGN and AC/DC separately (reviewers igz9, rWWk), and (3) ablations of the strategy for identifying affected nodes (reviewers igz9, rWWk). If you have any specific questions or concerns that could further strengthen your support for our work, we would be happy to address them.

---

### Official Review · Reviewer_igz9 · 2025-03-16

**Overall Recommendation:** 3

**Summary:**

The paper addresses the challenge of Corrective Unlearning in Graph Neural Networks (GNNs). While GNNs are widely used across various applications, their message-passing mechanism makes them vulnerable to adversarial manipulations and erroneous data, as errors can propagate throughout the graph. To mitigate this, the authors introduce Cognac, a novel method designed to unlearn the effects of manipulated data, even when only a small fraction (5%) of the manipulated set is identified. Cognac significantly outperforms existing approaches, restoring model performance to levels close to those achieved with fully corrected data. Moreover, it is 8× more efficient than retraining the model from scratch.

**Claims And Evidence:**

Generalizability to Diverse Manipulations and Tasks:
The paper primarily examines targeted binary class confusion attacks on edges and nodes within node classification tasks. The evaluation to a broader range of attack types and graph-related tasks is missing to validate Cognac's overall effectiveness and applicability.

**Essential References Not Discussed:**

no

**Experimental Designs Or Analyses:**

Evaluation Scope: The paper evaluates targeted binary class confusion attacks on both edges and nodes, covering key manipulation types. However, expanding the analysis to a wider range of attack scenarios could further strengthen the evaluation.

Identification of Affected Nodes: The paper employs a heuristic approach to identify affected nodes by inverting features and observing changes in output logits. While this method appears reasonable, a more rigorous validation against alternative approaches would enhance confidence in its effectiveness.

The paper does not explicitly include ablation studies to assess the contribution of individual components (CoGN and AC-DC). Incorporating such studies would strengthen the justification for the method’s design choices and provide deeper insights into their impact.

**Methods And Evaluation Criteria:**

yes

**Other Comments Or Suggestions:**

It would be great to have more experiment evaluation on the eally large-scale dataset.

**Other Strengths And Weaknesses:**

Strengths:
1. The paper provides a theoretical analysis of adversarial attacks on GNNs, specifically examining the effects of Interclass Confusion attacks on the Wasserstein-2 distance between class embedding distributions.
2. Cognac effectively mitigates the impact of manipulated data, even when only 5% of the manipulated set is identified, outperforming existing GNN unlearning methods.
3. The proposed method restores most of the performance of a strong oracle trained on fully corrected data, even surpassing retraining from scratch when the deletion set is excluded.
4. Computational Efficiency: Cognac is 8× more efficient than retraining and scales effectively to large datasets.


Weakness:
1. Scalability to Extremely Large Graphs: While Cognac demonstrates strong scalability to large datasets, its performance on massive graphs with billions of nodes, which are common in real-world applications like social networks, remains largely untested.
2. The study primarily focuses on node classification tasks and targeted binary class confusion attacks on edges and nodes, which may not fully capture all possible manipulation scenarios.
3. Dependence on Identified Manipulations: Although Cognac remains effective even when only 5% of manipulated entities are identified, its success still depends on having some prior knowledge of the manipulations.

**Questions For Authors:**

no

**Relation To Broader Scientific Literature:**

The paper builds upon the recently introduced problem of Corrective Machine Unlearning (https://arxiv.org/abs/2402.14015), which aims to remove the adverse effects of manipulated data while being agnostic to the type of manipulations. This approach operates with access to only a representative subset of the manipulated data for unlearning, making it more practical for real-world applications where full knowledge of manipulations is often unavailable.

**Theoretical Claims:**

Yes,
Theorem 3.1 analyzes the impact of Interclass Confusion attacks on class representations.

Lemma 3.2 establishes the locality of manipulation propagation in GNNs.

---

> ### Author Rebuttal · Authors · 2025-04-01
>
> We appreciate the reviewer’s recognition of our work, particularly noting our theoretical analysis of adversarial attacks on GNNs, the effective mitigation of data manipulation through Cognac, and its computational efficiency compared to retraining.
>
> ---
>
> ## Expanding the analysis to a wider range of attack scenarios
>
> Thank you for the suggestion! We've added preliminary experiments on feature poisoning backdoor attacks across 3 representative datasets, now covering all major poisoning types (label, graph structure, and feature). The results can be found at https://imgur.com/a/nJnsqWN.
>
> Our feature attack injects trigger a pattern into the feature vectors of select nodes, and assigning a fixed spurious label, and reduces accuracy on the target distribution. Despite not being the strongest possible attack (also addressed in the limitations section, L422-426), our implementation provides sufficient signal for evaluation - most unlearning methods struggle, while Cognac matches and even outperforms retraining performance. Retrain fails to recover accuracy on the victim class for the *Cora* dataset, while SCRUB fails to do so for both *Cora* and *CS*. GNNDelete and MEGU, the graph unlearning baselines, fail to remove the poison.
>
> ---
>
> ## Affected Nodes Sampling: Validation against alternative approaches
>
> Our ablations (Figure 8, Appendix E.2) show the top 10% of affected nodes using our sampling performs comparably to using all nodes. As per your suggestion, we compare our sampling with MEGU’s.
>
> The link to the table can be found at https://imgur.com/a/3HykVJ8. Our results on the Cora dataset indicate that our method outperforms MEGU’s approach. Our heuristic delivers over 25% higher $Acc_{aff}$ and is 8x faster (0.43s vs 0.05s for 3160 samples)
>
> Key difference: MEGU propagates features through adjacency matrices using cosine similarity to adaptively threshold selection, while we use actual model predictions with L1 norm.
>
> We would be happy to include any additional sampling strategies/influence functions in the final version.
>
> ---
>
> ## Method Ablations
>
> We appreciate your suggestion!
>
> We've added ablation studies showing the individual contributions of both components (CoGN and AC/DC). While our paper already reports AC/DC performance separately (Figure 3), we've now added results for CoGN too, at this link (https://imgur.com/a/mk6D9ly):
>
> Both components contribute significantly: **CoGN alone achieves only 34.7% forgetting on Amazon (compared to Cognac’s 82.9%) and 42.1% on Cora (vs. Cognac’s 75.5%)**, showing that it effectively moves affected nodes but lacks correction from labels. Conversely, **AC/DC achieves 76.2% forgetting on Amazon, far better than CoGN but still below Cognac**, confirming that the components are complementary. AC/DC weakens incorrect learning signals and preserves task-relevant representations, while CoGN steers affected nodes away from manipulated ones.
>
> ---
>
> ## Scalability to larger datasets
>
> We thank the reviewer for the suggestion. However, we note that Cognac is designed so that its computational complexity primarily scales with the **size of the deletion set, not the entire graph**. In real-world unlearning applications, the deletion set is typically a small fraction of the overall data, making our approach inherently scalable. Moreover, our experiments (Appendix D.2) demonstrate that even when faced with a relatively large deletion fraction, Cognac consistently outperforms baseline methods. This empirical evidence further confirms that our method can effectively handle large graphs, showing promise against scalability concerns even for massive real-world networks.
>
> ---
>
> ## Does Cognac’s success depend on having some prior knowledge of the manipulations?
>
> The unlearning algorithm is fully agnostic to the manipulation type (as we show in our experiments across label, graph structure, and the newly added feature attacks). The input to the algorithm is simply a representative subset of the manipulated data, without any knowledge about the manipulation itself. Furthermore, note that unlearning is only defined for a non-zero number of unlearning entities. Please see our discussion with reviewer rWWk for additional insights on this.
>
> ---
>
> Thanks! Your detailed feedback has helped us greatly improve the paper. We hope this increases your support for our work.

---

### Official Review · Reviewer_rWWk · 2025-03-19

**Overall Recommendation:** 3

**Summary:**

In this paper, the author proposes an unlearning algorithm, Cognac, to remove manipulated data from a well-trained GNN model. The approach first identifies sensitive neighbors that may be influenced by spurious entities and then mitigates these effects by aligning the embeddings of the selected neighbors with unaffected ones. Abundant experiments have been conducted to demonstrate the perrormance of the proposed method.

**Claims And Evidence:**

The claims are sound and figure 1 explains it by showing a concrete example. However, I have two questions:
1 In Section 2, the focus is on removing edges that compromise the homophily property of the graph. However, recent findings suggest that heterophilic GNNs can enhance performance, indicating that non-local neighbors can also contribute to node and edge classification tasks. Does this imply that solely targeting homophilic edges could mislead the unlearning algorithm?

2 In Section 3.1.1, the method selects affected nodes using a heuristic that measures the influence of manipulated data. I suggest providing a concrete example to clarify this approach.

3 In reality, we usually don't know the manipulated data set.

**Essential References Not Discussed:**

Yang, Tzu-Hsuan, and Cheng-Te Li. "When Contrastive Learning Meets Graph Unlearning: Graph Contrastive Unlearning for Link Prediction." In 2023 IEEE International Conference on Big Data (BigData), pp. 6025-6032. IEEE, 2023.

**Ethics Expertise Needed:**

["Responsible Research Practice (e.g., IRB, documentation, research ethics, participant consent)"]

**Experimental Designs Or Analyses:**

The experiments appear comprehensive, and their algorithm outperforms other approaches.

**Methods And Evaluation Criteria:**

The problem is well defined and algorithm is plausible. However, I have several questions:
1 In section 3.1.1, the method selects affected nodes influenced by the manipulated data. However, how can they ensure that these nodes exert a strong influence on other 'unaffected' data? For example, suppose node A is a 1-hop neighbor of the manipulated node N but has no other connections, while node B is a 2-hop neighbor of N but has multiple neighbors. Given k=1, does it make sense to select A over B?

2 I suggest them to introduce the procedure of spurious edge addition in their experiments.
3 They propose a contrastive loss in formula 2 by aliging the inner product of "affected links" with "unaffected links".  Could this potentially degrade the information preserved in the 'affected nodes'?

**Other Comments Or Suggestions:**

NA

**Other Strengths And Weaknesses:**

The idea is innovative. However, paper was a little hard to follow with too many avenues and experiments explored, it would have been more productive to reduce the scope of the paper. At the end the conclusion and future work needs to be a little more elaborate in terms of next steps.

The idea and approach is good, work is there just needs more organizing and scoping out.

**Questions For Authors:**

Could you provide the link of code?

**Relation To Broader Scientific Literature:**

NA

**Theoretical Claims:**

I suggest them to prove the soundess of the heuristic of selecting affected nodes in section 3.1.1.

---

> ### Author Rebuttal · Authors · 2025-04-01
>
> We appreciate the reviewer’s recognition of our comprehensive experiments and the soundness of our claims. We're also pleased that the reviewer found the idea behind Cognac innovative. We hope our clarifications below address your concerns.
>
> ---
>
> ## Affected Nodes Sampling
>
> _Can solely targeting homophilic edges mislead the unlearning algorithm?_
>
> Your question raises an important point. While the attack exploits the homophily due to the GNN’s inductive bias, the unlearning algorithm is fully agnostic to the manipulation type. It simply requires knowledge of a representative subset of the unlearning entities and selects affected nodes based on prediction changes, not graph structure properties, and can hence be adapted for heterophilic settings. Moreover, in L418-421, we acknowledge that our current work has not explored applications in heterophilic datasets yet.
>
> _1-hop vs 2-hop node selection_
>
> The heuristic selects nodes based on the magnitude of change in their output logits (∆χ) when features are inverted, and not just the hop distance. A high ∆χ indicates strong influence from manipulated nodes, even if the node is not a direct neighbor. By choosing the top k% based on the GNN forward pass, the method accounts implicitly for hop distance, ensuring that only nodes significantly affected are flagged for corrective unlearning.
>
> _Soundness of our heuristic selecting affected nodes_
>
> Lemma 3.2 shows that a manipulated node’s influence is confined to its n-hop neighborhood. Using feature inversion, we measure how changes in these nodes’ features affect their neighborhood. Our ablation studies (Figure 8, Appendix E.2) reveal that selecting the top 10% of nodes by change in output logits performs similarly to using the full n-hop subgraph, yet is 2× faster. Moreover, compared to MEGU’s sampling method, our heuristic delivers over 25% higher $Acc_{aff}$ and is 8× faster. Additional details are in our discussion with reviewer igz9 and will be included in the revised version.
>
> _Working Example_
>
> We will add an example of our affected nodes sampling to the updated version (link to figure: https://imgur.com/a/Kgzfl3E)
>
> ---
>
> ## Contrastive Loss and Information Preservation
>
> Thank you for pointing this out! Left unchecked, contrastive learning could indeed degrade information. That's why we perform gradient descent on the retain set, which contains these affected nodes, ensuring their information is preserved. As part of new experiments, we add an ablation showing that contrastive unlearning alone (CoGN) performs notably worse notably worse (upto 48.2% on label flip on Photos). More details can be found in our discussion with reviewer igz9 and in the following table: https://imgur.com/a/mk6D9ly. We will include this discussion explicitly in the revised version in Appendix E.
>
> ---
>
> ## Data Manipulations - Breadth and Discovery
>
> _Discovering manipulated data_
>
> We agree that addressing the difficulty in finding manipulated data is the motivation and main contribution of our work! [1] shows that prior works rely on the availability of all manipulated samples, which is unrealistic. Our method Cognac achieves better performance than theirs with as little as 5% of the manipulated set known. Note that unlearning is only defined for a non-zero number of unlearning entities. A small fraction of the manipulated data can be identified by manually investigating a small random subset of the data or using automated tools like [2]. We will address this in the revised version in Section 6.
>
> _Spurious edge addition_
>
> Thanks for noting that this would be a valuable setting!  This attack is indeed covered in our experiments, with details in Section 4.2.
>
> ---
>
> ## Paper Revision
>
> 1. **We’ll improve the writing of proof A.2** to make it more readable, and **expand the conclusion** with specific ideas for next steps.
> 2. **Include missing reference:** Thank you for pointing out the paper. We’ll add it to our related work section. We note that while both use contrastive methods, they use it for different objectives. Cognac performs contrastive learning directly on hidden representations of affected nodes while GCU improves upon GNNDelete by contrasting between Deleted Edge Consistency and Neighbourhood Influence to provide a more granular, graded removal of edge information rather than the binary deletion used in GNNDelete.
> 3. **Code is already present in the supplementary material:** All code and hyperparameters are in the supplementary materials zipfile; we'll include a deanonymized link in the final version.
>
> ---
>
> Thanks for your questions and suggestions. We hope our response increases your support for our work and are happy to discuss further!
>
> [1] Goel, Shashwat, et al. "Corrective Machine Unlearning." *Transactions on Machine Learning Research*.
>
> [2] Thyagarajan, Aditya, et al. "Identifying Incorrect Annotations in Multi-label Classification Data." *ICLR 2023 Workshop on Pitfalls of limited data and computation for Trustworthy ML*.

---

### Decision · Program_Chairs · 2025-05-01

**Decision:**

Accept (poster)

**Comment:**

This paper presents Cognac for corrective unlearning in Graph Neural Networks (GNNs). The approach seeks to remove the effects of manipulated data from a trained GNN model by identifying and mitigating the influence of spurious entities on sensitive neighbors. Cognac consists of two main components: one that repositions affected nodes away from manipulated entities and towards unaffected ones through contrastive learning, and another that uses gradient-based methods to induce unlearning for designated node sets. This requires the method to identify which nodes are manipulated, which may be difficult in practice (but assumed somewhat easy in the paper because of the clear influence they exert, which is OK). The authors provide some mathematical justification for the approach, as well as extensive experimental results across various benchmark datasets and GNN models.

Overall, the work is a good contribution. The reviewers were positive. I am recommend acceptance.